# Targeting Histone Deacetylases 6 in Dual-Target Therapy of Cancer

**DOI:** 10.3390/pharmaceutics15112581

**Published:** 2023-11-03

**Authors:** Milan Beljkas, Aleksandra Ilic, Alen Cebzan, Branko Radovic, Nemanja Djokovic, Dusan Ruzic, Katarina Nikolic, Slavica Oljacic

**Affiliations:** Faculty of Pharmacy, Department of Pharmaceutical Chemistry, University of Belgrade, Vojvode Stepe 450, 11221 Belgrade, Serbia; milan.beljkas@pharmacy.bg.ac.rs (M.B.); aleksandra.ilic@pharmacy.bg.ac.rs (A.I.); alen.cebzan@pharmacy.bg.ac.rs (A.C.); branko.radovic@pharmacy.bg.ac.rs (B.R.); nemanja.djokovic@pharmacy.bg.ac.rs (N.D.); dusan.ruzic@pharmacy.bg.ac.rs (D.R.)

**Keywords:** histone deacetylases, kinases, inhibitors, epigenetics, cancer, dual-target therapy, rational design

## Abstract

Histone deacetylases (HDACs) are the major regulators of the balance of acetylation of histone and non-histone proteins. In contrast to other HDAC isoforms, HDAC6 is mainly involved in maintaining the acetylation balance of many non-histone proteins. Therefore, the overexpression of HDAC6 is associated with tumorigenesis, invasion, migration, survival, apoptosis and growth of various malignancies. As a result, HDAC6 is considered a promising target for cancer treatment. However, none of selective HDAC6 inhibitors are in clinical use, mainly because of the low efficacy and high concentrations used to show anticancer properties, which may lead to off-target effects. Therefore, HDAC6 inhibitors with dual-target capabilities represent a new trend in cancer treatment, aiming to overcome the above problems. In this review, we summarize the advances in tumor treatment with dual-target HDAC6 inhibitors.

## 1. Introduction

Cancer is the second leading cause of death after cardiovascular disease. According to the World Health Organization, approximately 10 million deaths in 2020 were caused by cancer [1,2]. That is why novel treatments are so urgently needed. Until the early 2000s, alterations in a DNA sequence leading to the activation of oncogenes and loss of function of tumor suppressor genes were considered the main cause of tumorigenesis. However, it is now known that in addition to genetic alterations, epigenetic abnormalities also play an important role in tumor development and progression [1,3]. Post-translational modifications (PTMs) of histones such as methylation, acetylation, phosphorylation, ubiquitination, crotonylation and succinylation represent one of the most important epigenetic mechanisms [4].

One of the crucial PTMs affecting gene expression is histone acetylation, which occurs at lysine residues. It is catalyzed by a group of enzymes called histone acetyl transferases (HATs), while the reverse reaction, the removal of acetyl groups, is carried out by histone deacetylases (HDACs) [5,6]. The proper balance between the activities of HATs and HDACs is essential for normal cell function and state. 

In humans, there are 18 known isoforms of HDACs, which are classified into four classes based on their sequence similarity to yeast deacetylases, of which three classes contain zinc-dependent histone deacetylases (I, II, IV) and one class NAD+-dependent histone deacetylases (also known as sirtuins). HDAC6 belongs to class IIb and is predominantly localized in the cytoplasm [7,8]. Due to its localization, the main substrates of HDAC6 are not histones but other proteins in the cytoplasm such as α-tubulin, cortactin, heat shock protein 90 (Hsp90), Ku70, survivin [7,9,10,11,12,13]. In addition to deacetylase activity, HDAC6 interacts with several proteins of interest in oncology, including ubiquitin, tumor suppressor protein p53, epidermal growth factor receptor (EGFR) and c-Myc [14,15,16,17]. Overall, HDAC6 overactivity may contribute significantly to the initial steps of tumorigenesis and tumor progression through its influence on histone and non-histone proteins. Numerous studies have already demonstrated the overexpression of HDAC6 in tumors such as breast, liver, bladder, colorectal and neuroblastoma [17,18,19,20,21]. Therefore, it can be considered an important epigenetic target for tumor treatment.

Besides the distinctions in biological function and cellular localization of HDAC6 compared to other HDAC isoforms, HDAC6 also has a unique structure that can be exploited for the development of selective HDAC6 inhibitors (Figure 1). Unlike the other HDAC isoforms, HDAC6 contains two catalytic domains (CD1 and CD2), and its structure is characterized by the presence of a zinc finger domain with homology to ubiquitin-specific proteases that binds unanchored ubiquitin (ubiquitin-binding domain) [22,23]. In contrast to CD1, which is highly specific for substrates containing C-terminal acetyllysine residues (exo-acetyllysine peptide substrates), CD2 exhibits broader substrate specificity (exo- and endo-acetyllysine peptide substrates) [23]. Hai Y. and Christianson D. showed that a mutation in human CD2, but not in human CD1, leads to a greater than 400-fold reduction in the catalytic activity of HDAC6, indicating the importance of CD2 for the overall catalytic activity of HDAC6 [23]. Therefore, the binding modes and interactions with CD2 should be considered in the development of new HDAC6 inhibitors. Regarding the selectivity for the HDAC6 isoform among the other HDACs, previous studies have shown that the interactions between the CAP moiety of pan-HDAC inhibitors and H463 and P464 of the L1 loop are more important for binding to HDAC1, HDAC2 and HDAC3 than for binding to HDAC6, which can be used to design selective HDAC6 inhibitors [23]. In addition, S531 (part of the L2 loop) has been identified as a crucial amino acid residue for the recognition of α-tubulin [22]. Therefore, this part of HDAC6 should also be considered in the development of novel HDAC6 inhibitors. 

According to the classical pharmacophore model (Figure 2(IA)), the pharmacophore of any HDAC inhibitor (including selective HDAC6 inhibitors) consists of three parts: a ZBG (zinc-binding group), a CAP group and a linker [8]. The ZBG (such as hydroxamic acids, 2-aminobenzamides, thiols, carboxylic acids) coordinates zinc in a catalytic site, while the CAP group interacts with the surface of enzyme or amino acid residues near the outer domain of the active site. Structural modification of the CAP group may increase the selectivity of HDAC inhibitors [7]. However, some novel HDAC inhibitors cannot be explained by this model. Therefore, an extended pharmacophore model for HDAC inhibitors was recommended by Melesina J. and coauthors, which takes into account not only the parts of the inhibitors that interact with the main pocket (ZBG, linker and CAP group), as the classical model does, but also the parts that additionally target the subpockets [24]. The extended model compresses three additional parts compared to the classic model: the side-pocket targeting group (SP-group), the lower-pocket targeting group (LP-group) and the foot-pocket targeting group (FP-group) (Figure 2(IB)) [24].

To date, none of the selective HDAC6 inhibitors are in clinical use, but some of them (ricolinostat (ACY-1215), citarinostat (ACY-241) and KA2507 (Figure 2II) are in clinical trials for the therapy for different tumors such as relapsed/refractory lymphoid malignancies, metastatic breast cancer, melanoma, non-small cell lung cancer, etc. [25,26,27,28]. On other side, five pan-HDAC inhibitors are approved for use in humans, all in hematologic cancers: vorinostat (also known as SAHA), belinostat, panobinostat, chidamide (also known as tucidinostat) and romidepsin [29,30,31,32,33] (Figure 2III). Vorinostat is approved for the treatment of cutaneous T-cell lymphoma (CTCL), belinostat and chidamide for peripheral T-cell lymphoma (PTCL), panobinostat for multiple myeloma and romidepsin for both CTCL and PTCL.

Even though selective HDAC6 inhibitors have been developed, they have limited success in clinical trials as single-target therapy. The concerns have been raised about the use of HDAC6 inhibitors as single agents because their high concentrations show anticancer properties [34]. However, these high concentrations have been shown to impair their selectivity for HDAC6 which may lead to off-target effects. On the other hand, adverse effects and off-target toxicities limited the clinical use of pan-HDAC inhibitors due to non-selectivity. Given the above reasons, there are two trends in the development of HDAC inhibitors:(1)Increasing the selectivity toward one HDAC isoform (HDAC6) among others with the goal of reducing adverse effects;(2)Developing dual-target HDAC inhibitors in order to increase the efficacy and decrease the dose of HDAC6 inhibitors due to synergistic and additive effects.

Therefore, in this review, we focus on dual inhibitors, all targeting the epigenetic enzyme histone deacetylase 6 (HDAC6) and one of the following targets, such as phosphati-dylinositol 3′-kinases (PI3K), mammalian target of rapamycin (mTOR), bro-mo-domain-containing proteins 4 (BRD4), androgen receptor (AR), heat shock protein 90 (HSP90), tubulin, lysine-specific demethlylase 1 (LSD1), p-21 activated kinases 1 (PAK1), focal adhesion kinase (FAK), histone deacetylase 1 (HDAC1), histone deacetylase 3 (HDAC3) and histone deacetylase 8 (HDAC8). The main reasons for the rational design (Figure 3) of dual-target inhibitors of HDAC6 and the previously mentioned targets are as follows:(1)The synergistic effects of HDAC6 inhibitors in combination with PI3K, FAK inhibitors and microtubule stabilizers demonstrated in in vitro and/or in vivo studies [35,36,37];(2)The decreased efficacy of mTOR inhibitors and BRD4 inhibitors as single-target therapy due to the overactivity of HDAC6 [38,39];(3)The increased activity of AR and HSP90 due to the overexpression of HDAC6 [40,41];(4)The simultaneous targeting of non-histone proteins by inhibiting HDAC1, HDAC3, HDAC8, LSD1 and HDAC6 may show synergistic effects on cancer cell lines [42,43,44,45,46,47].

All of the dual inhibitors presented in this review are selective for the HDAC6 isoform among the other histone deacetylases. 

## 2. Multi-Target Therapy as an Approach in Cancer Treatment

Targeted cancer therapy emerged in the 1970s and 1980s following the identification of oncogenes, tumor suppressor genes and signaling pathways associated with cancer development [48,49,50]. This approach involves altering the activity of precise molecular targets that contribute to tumor growth and progression. The ideal molecular target should be specific and essential to the cancer cell, but this type of target has not yet been discovered [50,51,52]. For this reason, many studies are focused on investigating the signaling pathways involved in tumor development and progression to identify molecular targets and/or pathways that are to some extent essential and specific to cancer cells compared to normal tissues [50,53,54].

Because cancer is a multifactorial disease with a variety of pathogenetic mechanisms, targeting a single molecular target does not always lead to the desired outcome [55,56]. On the one hand, better efficacy and safety profile compared to chemotherapies are the main advantages, but a common problem in advanced disease is tumor resistance to a single-target therapy. Thanks to the additive and synergistic effects of a multi-target approach, the efficacy of cancer treatment is higher, while drug resistance is less frequent compared to single-target therapy [56,57]. Therefore, multi-target therapy is now one of the most commonly used approaches in tumor therapy and consists of two different strategies: the first is based on the combination of drugs acting on different targets (drug combination therapy), while the second involves the use of a single molecule that simultaneously affects the function of multiple targets—multi-targeting ligands [56]. Some drug combinations have already proven effective, e.g., dabrafenib (B-Raf serine-threonine kinase (BRAF) inhibitor) with trametinib (mitogen-activated extracellular signal-regulated kinase (MEK) inhibitor) in the treatment of metastatic melanoma with BRAF mutations; palbociclib (inhibitor of cyclin-dependent kinase 4 and 6, CDK4 and CDK6) with letrozole (aromatase inhibitor) in the treatment of advanced breast cancer, etc. [56,58]. The main problems with drug combinations are the difficulty in predicting toxicities, drug–drug interaction and metabolism and also complexity in clinical trials [51]. The main advantages of multi-targeting ligands compared to drug combinations, are the lower likelihood of drug–drug interactions, better patient adherence and a simpler pharmacokinetic and pharmacodynamic profile [57,58]. Some of the multi-target inhibitors have already been approved by the US Food and Drug Administration (FDA) for cancer treatment, such as entrectinib (multi-target inhibitor of tropomyosin, anaplastic lymphoma kinases (ALK) and receptor tyrosine kinase (ROS1)) [59].

The most important steps in the development of multi-targeting ligands are the validation of target combinations and the generation of lead structures [58]. Several strategies can be used to validate the target combinations: validation based on clinical observations, phenotypic screening or in silico technique. Two approaches are generally used for lead generation: screening and knowledge-based approach (also known as pharmacophore-based approach) [58,60]. The pharmacophore-based approach involves combining the pharmacophores of ligands for several targets into a single multi-targeting ligand. Depending on the overlap of pharmacophores, new multi-targeting ligands can be divided into three groups: multi-targeting ligands with linked, fused or merged pharmacophores [58,60]. Linked multi-targeting ligands are formed from the pharmacophores of the individual ligands connected by a linker. However, these multi-targeting ligands are usually bulky, which may affect their bioavailability. Moreover, in some cases, the linker may prevent the formation of key interactions between the multi-targeting ligand and targets. Fused multi-targeting ligands have partially overlapping pharmacophores, while merged multi-targeting ligands have the highest percentage of the overlap. Therefore, the molecular weight of merged multi-targeting ligands is the lowest, which may result in a better pharmacokinetic profile of these compounds compared to linked and fused multi-targeting ligands [58]. Regarding the screening approach, the most favorable strategy is focused screening where the class of compounds has already been shown to be active against one target of interest. Therefore, these compounds are screened against another target. This approach is commonly used for kinases type of targets [58,60]. 

The main approaches used in development of dual HDAC6 inhibitors are also discussed in this review. 

## 3. Dual HDAC6 Inhibitors

### 3.1. Dual HDAC6/PI3K Inhibitors

Phosphatidylinositol-3′-kinases (PI3K) are lipid kinases that play important roles in cancer initiation, growth, proliferation and survival as intracellular signal transducers. Abnormal activation of the phosphatidylinositol 3′-kinase/protein kinase B PI3K-AKT pathway is one of the critical factors for cancer cell survival and it is frequently dysregulated, leading to chemotherapy-resistant cancer cells [61,62].

To date, four PI3K inhibitors (idelalisib, selective for PI3Kδ [63]; copanlisib (BAY 80–6946), a pan-class I inhibitor [64]; duvelisib, a dual PI3Kδ and PI3Kγ [65] and alpelisib (PI3Kα) inhibitor [66]) have been approved by the FDA. However, the major problem is the limited efficacy of individual PI3K inhibitors due to the activation of alternative survival and growth pathways by tumor cells. Based on the results of previous studies that demonstrate synergistic effects between HDAC and PI3K inhibitors and considering the drawbacks of single therapy with HDAC or PI3K inhibitors (such as drug resistance), HDAC/PI3K dual-target inhibitors were developed [35,67].

**Fimepinostat (CUDC-907)** is a pan HDAC/PI3K inhibitor that is already in II phase of clinical trials for the treatment of metastatic and locally advanced thyroid cancer [68]. **Fimepinostat (CUDC-907)** was synthesized by integrating a hydroxamic acid (important for HDAC inhibition) into the structure core (morpholinopyrimidine) of two PI3K inhibitors (Figure 4I). **CUDC-907** showed inhibitory activity (IC_50_) of 19 nM, 54 nM, 39 nM and 311 nM for PI3Kα, PI3Kβ, PI3Kδ and PI3Kγ, respectively, and 1.7 nM, 5 nM, 1.8 nM, 27 nM, 2.8 nM and 5.4 nM for HDAC1, HDAC2, HDAC3, HDAC6, HDAC10 and HDAC11 [69]. Thus, **CUDC-907** can be considered as an HDAC and PI3K multi-target inhibitor [69]. However, the non-selectivity of this compound impacts its safety profile and promotes the undesirable tolerability [70]. Thus, the development of specific type of an HDAC/PI3K inhibitor may correlate with a better safety profile and tolerance. 

**LASSBio-2208** (**1**) is a dual HDAC6/PI3Kα inhibitor developed by modifying the structure of the HDAC6/HDAC8 inhibitor LASSBio-1911 (Figure 4II). After analyzing the main interactions between LASSBio-1911 and the HDAC6 crystal isoform (PDB: 5WGI) by molecular docking, Rodrigues D. and coauthors concluded that the 4-dimethylamino benzoyl moiety is solvent-exposed, so the modification of this part of the molecule will not have a major impact on the potency of the HDAC6 inhibitor [71]. At the same time, the structural changes in the 4-dimethylaminobenzoyl region may lead to PI3K inhibition. A pharmacophore-based approach is used for lead generation (fused pharmacophore model), which finally results in developing a dual HDAC6/PI3Kα inhibitor (**1**). Morpholine moiety of **1** established hydrogen bonds with the hinge region of PI3Kα (valine 851), while the hydroxyl group (position C-3 of phenyl ring) is already described as important for effective p110α inhibitor activity. The newly designed and synthesized **1** showed promising inhibitory activity at 15.3 nM for HDAC6, 67.6 nM for HDAC8, 46.3 nM for PI3Kα, 72.8 for PI3Kβ and 72.4 for PI3Kδ and was selected for further examinations [71]. 

Zhang Y. and coauthors presented **2** as a novel dual HDAC6/ PI3Kα inhibitor. **2** was designed by modifying the structure of the selective PI3Kα inhibitor (Alpelisib). Modifications were guided by the interactions observed in the cocrystal structure of PI3Kα (PDB: 4JPS) in a complex with alpelisib. The solvent-exposed part of alpelisib was identified (position C-4 of the L-prolinamide) (Figure 5I) and modified to incorporate the structural features of the HDAC6 inhibitor by applying a pharmacophore-based approach (Figure 4III). **2** showed high potency (IC_50_ = 2.9 nM and 26 nM against PI3Kα and HDAC6, respectively) and selectivity toward PI3Kα and HDAC6 isoforms. Molecular docking studies were performed with **2** and PI3Kα (PDB: 4JPS) and HDAC6 (PDB: 5EDU). Hydrogen bonds were formed between **2** and the amino acid residues of PI3Kα (Val851, Ser854 and Gln859), which were already described in the literature as important for PI3Kα inhibition. In addition, **2** coordinates the zinc ion in catalytic domain 2 of HDAC6 via a hydroxamic acid. This complex is additionally stabilized by hydrogen bonds with His610, His611 and Tyr782. Furthermore, the phenyl ring of **2** forms π-π interactions with Phe620 and Phe680 of HDAC6 [72]. 

Finally, another dual HDAC6/PI3Kδ inhibitor was developed by Li Z. and coauthors [70]. The structures of the approved PI3Kδ inhibitors have three structural features: the bicyclic heteroaryl that forms key interactions with the PI3Kδ-specific pocket, the hinge binder (HB) that interacts with the hinge region of the enzyme and the short linker that connects them. After analyzing the binding mode between the co cocrystal PI3Kδ and idelalisib (PI3Kδ inhibitor structurally similar to duvelisib) (Figure 5II), Zhi Li and coauthors concluded that the N-3 and C-4 positions of quinazolone are solvent-exposed domains that can be modified to achieve HDAC6 inhibition activity (Figure 4IV). The pharmacophore fusion strategy led to the discovery of **3**, which exhibited IC_50_ values of 2.3 nM and 13 nM against PI3Kδ and HDAC6, whereas the IC_50_ values for the positive controls, idelalisib and ACY-1215, were higher at 7 nM and 17 nM, respectively. In addition, **3** showed selectivity for PI3Kδ and HDAC6 subtypes (HDAC1, IC_50_ = 230 nM; HDAC8, IC_50_ > 5000 nM; HDAC11, IC_50_ = 1547 nM; PI3Kα, IC_50_ = 168.9 nM; PI3Kγ, IC_50_ = 30.5 nM) [70].

### 3.2. Dual HDAC6/mTOR Inhibitors

Mammalian target of rapamycin (mTOR) is a downstream component of the PI3K-Akt pathway and is involved in the regulation of numerous cellular processes such as cell growth and survival. mTOR is a serine/threonine kinase whose dysregulation has been found in human breast, prostate and kidney cancer. The overactivation of mTOR promotes tumor growth and progression. Therefore, this enzyme is an important target for cancer treatment [73]. Many mTOR inhibitors have been discovered, and some of them are already approved for cancer treatment, such as sirolimus, everolimus, temsirolimus and ridaforolimus [74,75,76,77]. However, despite the great expectations for the mTOR inhibitors, their efficacy in treating patients has been limited [38].

It has been reported that the expression of mTOR positively correlates with the acetylation of histones H3 and H4, whereas the inhibition of the Akt-mTOR axis rapidly decreases the level of aH3 (acetylated histone H3) and aH4 (acetylated histone H4). Deacetylation of histones is associated with cell proliferation, migration, invasion, etc. Therefore, a decrease in aH3 and aH4 levels may counteract the beneficial effects of mTOR inhibitors. Considering these relationships, simultaneous inhibition of mTOR and enzymes catalyzing deacetylation of histone H3—histone deacetylases—may improve the efficacy of mTOR inhibitors and prevent drug resistance [38]. Based on these findings, dual HDAC/mTOR inhibitors are being developed.

The structure of a selective HDAC6 inhibitor (**4**) discovered by Dahong Yao and coauthors was used to develop a novel dual HDAC6/mTOR inhibitor [73,78]. Although **4** has structural features important for binding to the active site of mTOR, it has no activity against mTOR. It was concluded that the length of the linker probably prevents the interactions between **4** and the active site of mTOR. Therefore, the proposed modifications in the structure of **4** were directed in two ways in order to design a potent dual HDAC6/mTOR l inhibitor. The first was the shortening of the linker, and the second was the removal of the methyl groups of the phenyl ring to facilitate the formation of a complex between Zn^2+^ and hydroxamic acid in order to increase the HDAC6 activity (Figure 6). Finally, **5** (a dual HDAC6/mTOR inhibitor) was synthesized and evaluated. It showed potent inhibitory activity against HDAC6 and mTOR with IC_50_ values of 56 nM and 133.7 nM, respectively, while the IC_50_ values for HDAC1, HDAC2 and HDAC3 were higher at 359.4 nM, 374.3 nM and 414.1 nM, respectively. In addition, a molecular docking study was performed demonstrating hydrogen bonds between the hydroxamic acid of **5** and the amino acid residues of mTOR (Val2240 and Trp2239). These interactions have already been reported in the literature to be important for binding of inhibitors for mTOR. In addition, the hydroxamic acid formed helate with Zn^2+^ within the catalytic domain of HDAC6, which was further stabilized by hydrogen bonds with His573 [73].

### 3.3. Dual HDAC6/BRD4 Inhibitors

The BET (bromodomain and extraterminal domain) protein family contains four members: BRD2, BRD3, BRD4 and the bromodomain testis-specific protein (BRDT). All bromodomains are readers of acetylated lysine residues on histones, but BRD4 is the best studied. BRD4 recognizes the acetylated lysine residues with N-terminal bromodomains and regulates transcription elongation with its C-terminal bromodomains by recruiting the positive transcription elongation factor b (pTEFb). The pTEFb is composed of CDK9 (cyclin-dependent kinase 9) and cyclin T and increases the expression of several growth-promoting genes. The BRD4 expression correlates with the overexpression of BCL2 and c-Myc. Thus, BRD4 is involved in the regulation of cancer cell processes such as cell cycle, proliferation, invasion, differentiation, growth and apoptosis [79,80]. Therefore, BRD4 inhibitors have been developed as small molecules that prevent interaction of BRD4 with acetyl-lysine and lead to decreased expression of genes that promote tumorigenesis [81].

The inhibition of BET proteins by the typical BET inhibitor JQ1 results in increasing the HDAC6 expression, which is associated with a reduction in the efficacy of JQ1. In light of this, Jennifer Carew and coauthors concluded that targeting BET-proteins and HDAC6 together could improve the antitumor properties of the BET-inhibitor against multiple myeloma, which was further confirmed by in vitro and in vivo studies [39]. The inhibition of HDAC6 by ricolinostat enhanced the apoptosis effects of JQ1 and the suppression of the c-Myc and BCL-2 expression. The synergistic effects were confirmed in LP-1 and OPM-2 multiple myeloma cells [39].

Inspired by the previous successes with combinations of HDAC6 and BET inhibitors, a new study was conducted by Chen J. and coauthors to design and test some dual HDAC6/BRD4 inhibitors [82]. Based on the binding mode of ABBV-744 and bromodomain 2 (BD2) of BRD2 (PDB: 6E6J), the solvent-oriented part of the molecule—the ethylamide region—was identified and modified in order to introduce the structural features of the HDAC6 inhibitor (Figure 7I). The pyrollo-pyridone feature was identified as critical for the BRD4 inhibition based on hydrogen bonding with the conserved Asn429 residue. A pharmacophore fusion strategy was employed, and **6** was developed as a dual HDAC6/BRD4 inhibitor (Figure 7II). It showed high potency in in vitro enzyme assays, with IC_50_ = 17.2 nM and 1.2 μM for HDAC6 and BRD4 (BD2), respectively. These results are better than those for the positive controls, SAHA (IC_50_ = 19.9 nM against HDAC6) and ABBV-744 (IC_50_ = 1.8 µM against BRD4 (BD2)). Moreover, selectivity for the HDAC6 isoform was confirmed by the HDAC isoform profiling (HDAC1, IC_50_ = 228.3 nM; HDAC3, IC_50_ = 161.2 nM; HDAC8, IC_50_ = 583 nM; HDAC11, IC_50_ = 2754.5 nM) [82]. 

### 3.4. Dual HDAC6/AR Inhibitors

The androgen receptor (AR) plays a central role in the development of prostate cancer [83]. Currently, there are a large number of antiandrogens that can be used to treat prostate cancer. However, drug resistance often occurs due to reactivation of AR, point mutations, ligand-independent activation pathways of AR, etc. As a result, castration-resistant prostate cancer (CRPC) may develop, which is very difficult to treat. This suggests that new treatment options for CRPC are needed.

Heat shock protein 90 (HSP90) plays a key role in androgen receptor activation. It forms a chaperone complex with AR and helps in establishing a ligand-binding conformation of AR [84]. Binding of the ligand leads to the importing of AR into the nucleus, where AR controls gene expression. Interestingly, HDAC6 may affect the activity of HSP90 by regulating its acetylation balance. In this way, the expression of HDAC6 could indirectly correlate with the activity of AR. It has been shown that the use of an HDAC6 inhibitor enhances HSP90 acetylation, leading to disruption of the HSP90-AR complex and degradation of AR [40,85].

Based on these findings, Maojun Zhou and coauthors developed a dual HDAC6/AR inhibitor—Zeta55 (**7**)—by using a merged-pharmacophore strategy [86] (Figure 8). In the structure of MDV3100 (AR inhibitor), the methyl group of the N-methylbenzamide moiety was replaced with a hydroxyl group, giving the molecule a zinc-binding group that is critical for the activity toward HDAC6. A molecular docking study was performed to reveal that **7** binds to HDAC6 and AR in a similar manner as HPOB (HDAC6 inhibitor) and bicalutamide (antiandrogen). **7** was evaluated by in vitro and in vivo assays. **7** strongly inhibited AR (IC_50_ = 0.63 µM) and selectively inhibited HDAC6 (IC_50_ = 0.98 µM) compared to HDAC1, HDAC2, HDAC3 and HDAC4. **7** had weaker activity against both targets compared to the positive controls (IC_50_ = 0.16 µM for vorinostat against HDAC6 and IC_50_ = 0.42 µM for MDV3100 against AR) [86].

Proliferation assays were performed with VCaP and LNCaP cells (AR positive prostate cancer cell lines), DU145 cells (AR negative prostate cancer cell line) and HEK293 cells (immortalized embryonic kidney cells), while MDV3100 and vorinostat were used as the positive controls. **7** showed a higher inhibitory effect on VCaP cells (IC_50_ = 2.47 µM) compared to MDV3100 (IC_50_ = 11.04 µM) and vorinostat (IC_50_ = 4.02 µM). **7** did not show significant inhibition in AR-negative cells (DU145 cells and HEK293 cells), suggesting that **7** achieves specific inhibition of AR-positive cells. Finally, **7** showed superior antitumor activity in mice with VCaP xenograft tumors compared to MDV3100 [86]. 

### 3.5. Dual HDAC6/HSP90 Inhibitors

The HSP90 protein family is highly conserved and widespread, serving as an ATP-dependent molecular chaperone involved in apoptosis, cell signaling, protein folding, degradation, cell cycle control and adaptive immunity. HSP90 has four isoforms: HSP90α and HSP90β (cytosolic), GRP94 (endoplasmic reticulum) and HSP75/TRAP-1 (mitochondrial). In cancer cells, HSP90 plays a crucial role in protecting mutated and overexpressed oncoproteins from misfolding and degradation, thereby ensuring their survival and promoting proliferation [87]. HSP90 inhibitors have been shown to lead to tumor shrinkage as well as differentiation and activation of apoptosis. Preclinical studies have shown that HSP90 inhibitors are effective in the treatment of castration-resistant prostate cancer, breast cancer, colon cancer, leukemia and melanoma [88]. 

HDAC6 regulates HSP90 function by deacetylation, thereby affecting the stability of HSP90 client proteins. The inhibition of HDAC6 leads to acetylation of HSP90, which decreases the binding of client proteins to HSP90, resulting in decreased activity and degradation of client proteins. Some studies have shown that the inhibition of HDAC6 can directly affect HSP90 fragmentation. In addition, it has been shown that HDAC6 can be one of the client proteins of HSP90 that regulates its degradation [41,89,90]. Given the interdependence of HDAC6 and HSP90, along with their recognized roles in critical cellular processes, they present themselves as ideal candidates for a multi-target approach to cancer treatment [41]. 

Rita O. and coauthors presented several structures of dual HDAC6/HSP90 inhibitors developed using the fused pharmacophore strategy. **8** and **9** were developed by modifying the structure of a previously known HDAC6 inhibitor (Figure 9I) [89,91]. The benzenesulfonyl group of the HDAC6 inhibitors was replaced by 4-isopropyl resorcinol, which is an essential structural feature for HSP90 inhibition and interacts with the ATP-binding site of the HSP90 proteins. In addition, the length of the linker was modified to investigate its effect on HDAC6 and HSP90 inhibition. Finally, **8** and **9** were effective against both targets: IC_50_ = 1.15 nM and IC_50_ = 4.32 nM for HDAC6 and IC_50_ = 46.3 nM and IC_50_ = 46.8 nM for HSP90, respectively. Compared to the positive controls, BIIB021 and trichostatin A (IC_50_ values of 65.7 nM and 2.6 nM, respectively), both compounds had lower IC_50_ values for both targets [89,91]. **9** showed its efficacy in in vivo studies. Human NSCLC H1975 xenograft model was used for evaluation of **9**, and it showed greater activity (higher percent of tumor growth inhibition) compared to afatinib (positive control) [89].

In addition, Tung-Yun and coauthors revealed one more dual HDAC6/HSP90 inhibitor (**10**) with a hydroxamic acid and resorcinol moiety as key structural features for the inhibition of HDAC6 and HSP90 [92] (Figure 9II). A biological evaluation showed that **10** is effective and selective for HDAC6 among other HDAC isoforms (HDAC1 and HDAC3), with an IC_50_ value of 4.56 nM, which is higher than the IC_50_ value for the positive control, trichostatin A (3.34 nM), and an IC_50_ value of 52 nM showing good efficacy against HSP90 (also higher compared to the positive control, geldanamycin, IC_50_ = 22.4 nM) [92]. Moreover, the antitumor properties of **10** were confirmed in an in vivo study in which it inhibited the growth of colon tumors in mice without causing significant toxicity [92].

### 3.6. Dual HDAC6/Tubulin Inhibitors

Alpha and beta tubulin heterodimers polymerize to form microtubules, essential components of the cytoskeleton of eukaryotic cell. These microtubules play a pivotal role in maintaining cellular architecture, cell division, intracellular transport, signaling and motility [93,94]. There are nine different isotypes of alpha and beta tubulin. The varied expression and post-translational modifications of these tubulin isotypes affect microtubule structure, dynamics and function in complex ways [93]. Perturbation or overexpression of these proteins has been associated with cancer. In particular, specific tubulin isotypes have been identified in cancer tissues, where they regulate tumor progression, metastasis, drug resistance and invasiveness [93].

The importance of microtubules as anticancer drug targets has been recognized for decades. Based on their mechanisms of action, several drug categories have emerged, including inhibitors of tubulin polymerization that interact with the colchicine binding site, such as Vinca alkaloids (vincristine, vinorelbine, vinblastine) and epothilones (ixabepilone) [95,96]. Conversely, depolymerization inhibitors act at the taxane binding site (cabazitaxel, paclitaxel, docetaxel). Notably, agents that bind to the colchicine binding site have lower susceptibility to resistance development [94]. Vincristine and vinblastine, which destabilize microtubules, are used in the treatment of Hodgkin’s lymphoma, while microtubule-stabilizing agents such as paclitaxel and docetaxel are used in solid tumors, including breast, lung, ovarian and prostate cancers [94,97]. 

Taking into account that HDAC6 controls microtubule dynamics by deacetylating alpha-tubulin, it can be concluded that the combination of HDAC6 inhibitors and microtubule stabilizing drugs could exert synergistic anticancer effects [95,96]. Considering the results of a previous study which has already demonstrated synergistic effects between paclitaxel and a selective HDAC6 inhibitor (citarinostat) [37], there is a rational case for the development of dual-target inhibitors.

A series of N1-substituted 3-aroylindoles were designed and synthesized by H. Y. Lee and coauthors with the aim of targeting tubulin and HDAC6 activity [95]. These compounds were designed by modifying the structure of the already known tubulin-assembly inhibitor SCB01A, considering the results of a previous study (SAR) that showed that the N1 position of SCB01A can be changed without significant effect on the inhibitory and antiproliferative activity against tubulin. In view of this conclusion, H. Y. Lee and coauthors introduced different hydroxamic acid moieties at the N1 position to obtain dual HDAC6/tubulin inhibitors (Figure 10I). In this way, **11** and **12** (Figure 10I) were designed, synthesized and further evaluated by biological assays [95,96]. They demonstrated good HDAC6 inhibition activity with IC_50_ values of 64.5 nM and 275.35 nM, respectively, and remarkable selectivity compared to other HDAC isoforms (HDAC1, HDAC2 and HDAC8). The effects of the two compounds on microtubule dynamics were confirmed by an in vitro tubulin polymerization assay. Compared to the positive control, SAHA (IC_50_ (HDAC6) = 72.34 nM with onefold and fourfold increases in selectivity for HDAC6 over HDAC1 and HDAC2, respectively), **11** is a stronger HDAC6 inhibitor and has better selectivity for HDAC6 among the other HDAC isoforms, while **12** (MPT0B451) has lower activity but better selectivity [95,96]. Moreover, their antitumor properties have been confirmed in in vivo studies. **11** inhibited tumor growth in two different in vivo models, a human prostate PC3 xenograft model and an RPMI-8226 cancer cell xenograft model, resulting in tumor growth inhibition (TGI) of 68.5% and 58.2%, respectively, with no change in the weight of the experimental animals [95]. To further evaluate the antitumor activity of **12**, human leukemia cells (HL-60) and human prostate cancer cells (PC-3) were used in mouse xenograft models, resulting in tumor growth inhibition (TGI) of 40.9% and 31.1% in mice with transplanted HL-60 and PC-3 cells, respectively [96].

Kumar K. and coauthors developed a group of dual HDAC6/tubulin inhibitors with a pharmacophore based on the indanone core as a backbone, which was previously used for the development of microtubule destabilizers. Indanone was modified in the C-2 position to introduce the structural features of HDAC6 inhibitors and provide HDAC6 inhibition activity (Figure 10II) [98]. The new HDAC6/tubulin inhibitor (**13**) was designed and synthesized by these authors. Its ability to inhibit HDAC6 in a selective manner was investigated using HeLa nuclear extracts. The residual HDAC activity in the HeLa nuclear extract containing two classes of HDACs (class I and II) in the presence of 20 µM of **13** was 23%, which is comparable to that of tubastatin A and trichostatin A (35% and 17%, respectively), while the residual HDAC6 activity at 20 µM was 3%. This indicates that **13** is an effective and highly selective HDAC6 inhibitor. The effects of **13** on stabilizing microtubule dynamics were confirmed by a tubulin kinetics study and confocal microscopy [98].

Finally, another dual HDAC6/tubulin inhibitor was described by Wang F. and coauthors [99]. They used a screening approach for lead generation. First, **14** (Figure 10III) was described as a selective HDAC6 inhibitor with an IC_50_ = 17 nM (a 25-fold and 200-fold increases in selectivity for HDAC6 over HDAC1 and HDAC8), whose activity against solid tumors was better than that of the previously reported selective HDAC6 inhibitor (ricolinostat) and pan-HDAC inhibitor (SAHA) [100]. Therefore, the authors hypothesized that the inhibition of HDAC6 is not the only mechanism of action of **14**. This hypothesis was confirmed by using an HDAC6 knockout cell line, in which **14** still exhibited antitumor properties that were definitely independent of HDAC6. Therefore, further studies were conducted and showed that **14** also targets microtubules, independently of HDAC6, contributing to its antitumor effect. In addition to the results of the in vitro studies, **14** showed greater efficacy compared to ricolinostat in HBL-1, HCT-116 and A2780s xenograft models [99].

In summary, the intricate interplay of tubulin isotypes, microtubule dynamics and their regulatory elements in cancer progression underscores their significance as therapeutic targets [95,101]. The versatile properties of the aforementioned dual inhibitors, including their dual-binding domains and potent HDAC6 inhibition, highlight their potential as a novel avenue for cancer therapy [95].

### 3.7. Dual HDAC6/LSD1 Inhibitors

Lysine-specific demethlylase 1 (LSD1, also known as AOF2 or KDM1A) belongs to the family of flavin-dependent lysine-specific demethylases (LSDs) [102]. It is the first histone demethylase discovered in humans and is now an important epigenetic target that demethylates only the mono- and dimethylated lysine 4 (H3K4) or lysine 9 of histone 3 (H3K9) [102,103]. In addition to histone targets, this enzyme plays an important role in the balance of methylation of non-histone proteins such as the tumor suppressor protein p53, myosin phosphatase target subunit 1 (MYPT1), the SRY box 2 (sex determining region Y), DNA methyltransferase 1 (DNMT1), E2F transcription factor 1 (E2F1), signal transducer and activator of transcription 3 (STAT3) and hypoxia-inducible factor 1α (HIF1α) [47,103,104]. The overexpression of LSD1 has been found in various human cancers such as gastric cancer, prostate cancer, acute myeloid leukemia, breast cancer, liver cancer, lung cancer, colorectal cancer, pancreatic cancer, neuroblastoma and many more [103,105]. The overexpression of LSD1 is closely related to differentiation, proliferation, migration, invasion and poor prognosis of tumors [47]. The inhibition of LSD1 by small molecules is associated with blocking cell growth and migration, as well as re-express the epigenetically silenced tumor-suppressor genes suggesting that LSD1 inhibitors may represent an important therapeutic approach for cancer treatment [104,105,106]. To date, several LSD1 inhibitors are already in various phases of clinical trials (tranylcypromine, ORY-1001, ORY-2001, GSK-2879552, INCB059872, IMG-7289, TAK418, CC-90011 and SP2577), especially for the treatment of acute myeloid leukemia and small lung cancer cells [107,108]. Previous studies have shown that there is an interaction between LSD1 and HDACs, as both are part of the CoREST complex, which is associated with silencing the gene expression and plays an important role in cancer cell survival and proliferation [109]. Based on these findings, numerous research studies show that combined inhibition of LSD1 and HDACs is more effective than the inhibition of a single enzyme in stopping the growth and migration of various tumors, including breast cancer, AML and glioblastoma. In this way, one of the most potent pan-HDAC/LSD1 inhibitors, corin, was discovered by Kalin J. and coauthors [109] (Figure 11I) whose main disadvantage is non-selectivity that may correlate with its adverse effects. Therefore, some of the studies now focus on the development of dual HDAC6/LSD1 inhibitors. LSD1 and HDAC6 affect the function of numerous non-histone proteins, some of which are the target of both enzymes [47,103,104]. In addition to the effects of LSD1 and HDAC6 on non-histone proteins, recent studies have shown that HDAC6, together with the CoREST complex, may play an important role in the estrogen receptor gene expression (ER) in breast cancer [110]. Based on these findings, Gajendran C. and coauthors and Sadhu N. and coauthors reported the discovery of a new LSD1 and selective HDAC6 inhibitor—**15** (Figure 11I) [111,112]. 

In biochemical assays **15** has shown good potency against both enzymes (LSD1 and HDAC6) with IC_50_ values of 6 nM and 48 nM, respectively. In terms of selectivity, this inhibitor has a greater than seventyfold increase in selectivity for HDAC6 over HDAC1 and a twofold increase in potency against HDAC6 compared to that against HDAC8. Moreover, it shows potent antiproliferative activity in the MM. 1S multiple myeloma cell line with an EC_50_ value of 2 nM [112]. The good efficacy of the **15** was also confirmed by in vivo studies in various xenograft tumor models. **15** showed 76% and 91% tumor growth inhibition (TGI) in xenograft mouse models of erythroleukemia (HEL92.1.7) at a dose of 25 mg/kg and 50 mg/kg, respectively. The combination therapy showed better efficacy than single therapy in the treatment of multiple myeloma MM1.s xenograft model. The single therapy (**15**) showed a TGI of 23% at a dose of 12.5 mg/kg, while the combination with bortezomib or pomalidomide showed strong inhibition of tumor growth of 82% and 60%, respectively. Finally, the **15** as a single agent showed a TGI of 50% in the treatment of CT-26 mouse colon carcinoma, while its combination with the anti-PD-L1 antibody resulted in an increase in efficacy to a TGI of 76% [111].

In addition, Bulut I. and coauthors revealed a potent LSD1/HDAC6 inhibitor—**16**—developed using the fused pharmacophore strategy (Figure 11II) [113]. GSK2879552 is a clinical candidate that is highly potent for LSD1. Bulut I. and coauthors hypothesized that replacing the carboxylic acid with a hydroxamic acid in GSK2879552 would not affect the LSD1 inhibition activity but could provide additional targeting for HDAC6. In this way, **16** was designed, synthesized and evaluated by enzyme assays. It demonstrated a great potency against LSD1 and HDAC6 with IC_50_ values of 0.54 µM and 0.11 µM, respectively. Compared to the clinical candidate GSK2879552, **16** had a threefold increase in potency while also exhibiting great isoform selectivity—greater than a thirtyfold increase for HDAC6/HDAC1, which is significantly higher compared to the clinical candidate ricolinostat (about a tenfold increase) [113].

### 3.8. Dual HDAC6/1, HDAC6/3, HDAC6/8

Histone deacetylases 1, 3 and 8 (HDAC1, HDAC3, HDAC8) all belong to class I of histone deacetylases and are predominantly localized in the nucleus [8]. In contrast to HDAC6, the main substrates of these enzymes are histones, but they also have some other targets. Because this review summarizes dual isoform-selective HDAC inhibitors without other targets, and the following compounds are not derived from the fusion or combination of two or more pharmacophores. Instead, they are the result of modifications to the structures of already known HDAC inhibitor pharmacophores (the CAP group, the linker and the ZBG), whether they were developed intentionally or by serendipity, or whether they simply exhibit dual inhibitory potential.

#### 3.8.1. Dual HDAC6/1 Inhibitors

In addition to histones, HDAC1 also deacetylates several other substrates, namely tumor suppressor protein p53, LSD1 (described earlier in this paper), transcription factor E2F1 and Eg5 [42,114,115,116]. The overexpression of HDAC1 and its contribution to tumorigenesis has been found in various tumors, such as gastric, prostate, liver, breast (as well as HDAC6 and HDAC8), colorectal and renal carcinoma (alongside with HDAC6) [117,118,119,120,121,122,123]. A meta-analysis concluded that the HDAC1 expression can serve as a diagnostic and prognostic factor for lung cancer [124]. Silencing of HDAC1 in ovarian cancer cells was found to overcome resistance to cisplatin [125], and silencing of both HDAC1 and HDAC6 was found to enhance cytarabine-induced apoptosis of acute myeloid leukemia (AML) cells induced by cytarabine [126]. 

Cheng C. and coauthors reported the discovery of a class of quinazoline-based HDAC1/6 inhibitors, developed from an in-house selective HDAC6 inhibitor [127]. The most potent inhibitor from this class is **17** (Figure 12), which consists of a hydroxamate as the ZBG, a 4-atom alkyl linker and a phenyl group linked to a quinazoline as the CAP group via a nitrogen atom. **17** displayed good potency against HDAC1 and HDAC6 isoforms with IC_50_ = 31.1 nM and IC_50_ = 16.15 nM, respectively. **17** showed remarkable antiproliferative activity against eight cancer cell lines (myeloma U266 and RPMI8226 cells, cervical cancer Hela cells, liver cancer HepG2 cells, lung cancer H1975 and H460 cells and breast cancer M-M-231 and MCF-7 cells), with IC_50_ values ranging from 0.1 nM to 3.50 nM and the best results in myeloma cell lines [127].

A pyridone-based class of HDAC inhibitors was developed by Cho and coauthors from their in-house lactam-based HDAC inhibitors, with **18** (Figure 12) showing the best selectivity and activity toward HDAC1 and HDAC6 isoforms with IC_50_ values of 19.4 nM and 2.46 nM, respectively. Its inhibitory activity for HDAC1 is comparable to that of the positive control, vorinostat (IC_50_ = 11.4 nM). **18** also showed better potency against HDAC6 (IC_50_ = 16.01 nM) compared to vorinostat. The main modification was in the linker domain, while the hydroxamate was retained as a ZBG and simple groups (such as phenyl and naphthyl) were replaced for CAP group. For better metabolic stability, a conjugation from the hydroxamic acid to the pyridone core was introduced into the linker, and the pyridone core was connected to the CAP group (2-naphthyl in the case of **18**) via an alkyl chain. The growth inhibition effect of this class on cancer cells was studied in six cancer cell lines (breast cancer MDA-MB-231 cells, renal cancer ACHN cells, colon cancer HCT-15 cells, prostate cancer PC-3 cells, gastric cancer NUGC-3 cells and non-small cell lung cancer NCI-H23 cells). **18** showed the best inhibitory activity with GI_50_ values between 0.14 µM and 0.38 µM [128].

Some studies have reported a novel, harmane-based class of HDAC1/6 inhibitors [129,130,131,132]. The most potent inhibitors from each study are designated as **19** (**HBC**), **20**, **21** (**CHC**) and **22** (Figure 12). All series have a bulky 1-phenylharmane as the CAP group and a hydroxamic acid as the ZBG. The hydroxamate is either directly connected or conjugated to a phenyl group in the linker, which is connected to the harmane core in various ways. All these compounds demonstrated great inhibitory activity for HDAC1 (IC_50_ values between 1.3 nM and 29 nM) and HDAC6 (IC_50_ values between 2.6 nM and 13 nM). They were tested on four different HDAC isoforms (HDAC1, HDAC3, HDAC6, HDAC8) and showed good selectivity for HDAC1 and HDAC6 among the other isoforms. Also, these compounds have been shown to induce apoptosis of various cancer cells. **19** and **21** have been shown to significantly reduce the size of hepatic tumors in in vivo tests [129,131].

#### 3.8.2. Dual HDAC6/3 Inhibitors

In addition to HDAC1, HDAC3 also deacetylates some non-histone substrates, including the NF-kB protein RelA, p53, myocyte enhancer factor 2 (MEF2), p300/CBP (E1A binding protein p300/CREB-binding protein), CDK9 [133,134,135,136]. Upregulation and involvement of HDAC3 has been found in renal, colon and breast cancers, as well as in leukemia [137,138,139,140]. Besides that, selective inhibition of HDAC3 has also shown cytotoxic effects on a melanoma cell line [141]. Some studies suggest that simultaneously affecting HDAC3 and HDAC6 via the survivin and tubulin axes may have a synergistic effect on the treatment of cancer cells [43,44]. A novel hybrid of **vorinostat** and **glycyrrhetinic acid** has been shown to reduce protein levels of HDAC3 and HDAC6 that induce death of PC-3 and HL-60 cells [142].

With a little bit of serendipity, Soumyanarayanan and coauthors discovered a novel selective HDAC3/6 inhibitor, **23** (Figure 13), during the development of dual HDAC and G9a inhibitors [143]. The strategy behind this design was to fuse the structure of vorinostat with the aniline derivative of BIX01294, a G9a (histonemethyl transferase) inhibitor. The two structures overlap at the phenyl group of the CAP group of vorinostat and the aniline group of the BIX01294 derivative. **23**, as an HDAC inhibitor, has the same ZBG and linker as vorinostat, while the phenyl group connected to the entire pharmacophore of the G9a inhibitor is considered to be the CAP group. **23** has an IC_50_ value of 34 nM for HDAC3 and of 2.6 nM for HDAC6 and exhibits a 300–3000-fold increase in selectivity over other isoforms [143].

#### 3.8.3. Dual HDAC6/8 Inhibitors

HDAC8, similar to the previously discussed isoforms, targets cortactin and contributes to the regulation of the p53 expression [46,144]. A notable physiological target of HDAC8 is the structural maintenance of chromosomes 3 protein (SMC3), a protein that holds two sister chromatids together during the progression of the cell cycle [145]. HDAC8 also deacetylates and enhances the transcriptional function of estrogen-related receptor α (ERRα) [146]. The role of HDAC8 has been demonstrated in various malignancies such as acute myeloid leukemia, neuroblastoma, hepatocellular, breast and colon cancers [147,148,149,150,151]. The knock-out, inhibition and degradation of HDAC8 by proteolysis targeting chimeras (PROTACs) have all shown positive results in targeting cancer cell lines [150,151,152,153,154].

The overexpression of HDAC8 along with HDAC1 and HDAC6 has been shown to promote invasion of MDA-MB-231 and MCF-7 breast cancer cell lines [121]. Vanaja and coauthors found that HDAC8 targets tubulin in HeLa cervical cancer cells and that inhibition or silencing of HDAC8 impedes the migration of these cells [45]. Based on these findings, the combined targeting of HDAC6 and HDAC8 may lead to more comprehensive inhibition of tubulin deacetylation. The combination of a selective HDAC8 inhibitor PCI-34051 with a selective HDAC6 inhibitor citarinostat synergistically suppressed migration and induced apoptosis in p53 wild-type ovarian cancer cells [155]. Considering the results of aforementioned studies, a dual-target HDAC6/HDAC8 inhibitor was developed. 

The first series of dual HDAC6/8 inhibitors was developed in 2013 by altering the structure of selective CAP-less HDAC6 inhibitors, which yielded isophthalamide derivatives, with **24** (Figure 14) showing the best activity for HDAC6 and HDAC8 at IC_50_ values of 21 nM and 37 nM, respectively. An interesting remark about these compounds is that the phenyl linker is meta-substituted, unlike selective HDAC6 inhibitors which have para-substituted linkers. The explanation for this lies in the orientation of these substituents toward the solvent; thus, their binding is not of great importance while they interact with the specific pocket of HDAC8. The inhibitory effect of **F24** was also confirmed in biochemical assays in HeLa cells [156].

Rodrigues and coauthors developed a new series of HDAC6/8 inhibitors by modifying the structure of trichostatin A (TSA), a pan-HDAC inhibitor [157]. The CAP group and ZBG of TSA remained unchanged, while a bulkier p-substituted N-acylhydrazone structure was introduced as the linker, further supporting the assumption of the selectivity of bulkier inhibitors. Overall, four compounds showed favorable inhibitory activity, with **25** (Figure 14) having the strongest effect with IC_50_ values of 97 nM against HDAC6 and of 54 nM against HDAC8.

Tang and coauthors reported a class of novel aminotetralin HDAC6/8 inhibitors, of which **26** (Figure 14) was the most potent with IC_50_ values of 50 nM (HDAC6) and 80 nM (HDAC8). Also in this class, the hydroxamic acid was retained as the ZBG, while aminotetralin was introduced as the linker, with variations in a CAP group (**26** has a pyridine linked to a pyrimidine as a CAP group). It is worth noting that the R enantiomers have higher potency, while the S enantiomers have lower activity toward HDAC8, which was verified by molecular docking. **26** also moderately inhibited the growth of myeloma cell line NCI-H929 with an EC_50_ value of 7.7 µM, compared to an EC_50_ value of 0.8 µM for vorinostat [158].

Negmeldin and coauthors conducted a series of studies with vorinostat modifications as HDAC6/8 inhibitors [159,160,161]. In these studies, the authors only modified the linker of vorinostat, while the CAP group and the ZBG were retained. The linker was modified by adding alkyl or aryl substituents to the carbon atoms at positions C2, C4 and C5. The C4 analogs were the most potent, followed by the C5 analogs, while the C2 analogs were the least potent. **27** (Figure 14), the R-enantiomer of the benzyl C4 vorinostat analog, was the most potent of the designed inhibitors with IC_50_ values of 48 nM (HDAC6) and 27 nM (HDAC8), while the S-enantiomer was less potent. The racemate of **27** and its enantiomer showed an EC_50_ value of 28 µM in U937 leukemia cells.

In 2022, a novel class of bulky HDAC6/8 inhibitors was reported, in which azetidin-2-one is connected to piperidine as a CAP group. In this class, hydroxamate was retained as the ZBG while using a phenyl linker variation. The most potent inhibitor in this class is **28** (Figure 14) with two phenyl groups in the *trans* position on two adjacent carbon atoms of the azetidin-2-one. It showed good inhibitory activity against HDAC6 and HDAC8 with IC_50_ values of 21 nM and 42 nM, respectively. This compound, alongside with two others from this class reduced the proliferation of leukemia U937 and colorectal HCT116 cells. The inhibitory profile of these compounds can be extended to other HDAC isoforms to fully evaluate their selectivity [162].

### 3.9. Dual HDAC6/PAK1 Inhibitors

The p21-activated kinase 1 (PAK1) belong to the serine-threonine kinase family. The overexpression of PAK1 or amplification of the PAK1 gene has been associated with tumors such as breast cancer, ovarian cancer, colorectal cancer, hepatocellular carcinoma and many others. PAK1 has been shown to play a role in cancer initiation and progression by regulating cancer growth, angiogenesis, metastasis, survival, tumor immunity and metabolism and drug resistance [163].

Zhang Y. and coauthors hypothesized that dual inhibition of HDAC6 and PAK1 could have strong implications for tumor treatment by simultaneously targeting oncogenic metabolic pathways and epigenetic modification [164]. Therefore, they reported a new dual HDAC6 inhibitor—a dual HDAC6/PAK1 inhibitor (**29**). The molecular docking study of the known PAK1 inhibitor showed that the aminopyrimidine moiety is the most important part of PAK1 inhibition due to hydrogen bonding to Leu3347 (hinge region of the enzyme), while the phenyl group interacts with the hydrophobic pocket of PAK1. At the same time, hexanolactam was identified as a solvent-exposed region that is not important for PAK1 inhibition. Therefore, this part of the molecule is suitable for modification to achieve the inhibition of HDAC6. As a result of these findings, **29** was designed and synthesized (Figure 15). It strongly inhibited HDAC6 and PAK1 with IC_50_ values of 38.23 nM and 13.62 nM, respectively. Moreover, it showed selectivity for HDAC6 and PAK1 among other HDAC and PAK isoforms (HDAC1,2,3,8,10 and PAK2,3). Besides the efficacy in in vitro studies, **29** demonstrated promising therapeutic potential for triple-negative breast cancer in vivo (MDA-MB-231 xenograft zebrafish and nude mouse tumor models) [164].

### 3.10. Dual HDAC6/FAK Inhibitors

Focal adhesion kinase (FAK) is a non-receptor tyrosine kinase involved in various cellular processes such as survival, proliferation, adhesion, migration, angiogenesis, stem cell formation and cytokine expression. Some FAK inhibitors are already in clinical trials for the treatment of solid tumors [165]. Dawson C.J. and coauthors performed a high-content chemical–genetic phenotype screening to identify drugs that might have synergistic effects with FAK inhibitors in order to resolve the disadvantages of single-target therapy. The results of this study showed that HDAC inhibitors in synergy with FAK inhibitors could induce apoptosis and stop cancer cell proliferation in several cancer lines, while this combination inhibited tumor growth in vivo [36]. Based on these findings, Song J. and coauthors discovered a new dual HDAC6/FAK inhibitor (**30**) [166].

Considering the binding mode of the already known FAK inhibitor (TAE226) and the kinase domain of FAK, the two crucial structural features for FAK inhibition were identified: the carbamoyl group and the pyrimidine ring (Figure 16I). Therefore, these two groups were included in the structure of an HDAC inhibitor (SAHA) as part of the CAP group. As a result of the pharmacophore-based approach, **30** was designed and synthesized (Figure 16II). It demonstrated excellent inhibitory activity against HDAC6 (IC_50_ = 16 nM), which is very similar to that of the positive control, SAHA (IC_50_ = 17 nM). However, compared to SAHA, **30** showed greater selectivity for HDAC6 among the other HDAC isoforms (with about a 33-fold increase in selectivity for HDAC6/HDAC1, about a 39-fold increase in that for HDAC6/HDAC2 and a 65-fold increase in that for HDAC6/HDAC3). **30** displayed good inhibitory activity against FAK with an IC_50_ value of 132 nM, which was much higher than that of the positive control, TAE226, with an IC_50_ value of 7 nM. The antiproliferative activity of **30** against several tumor cell lines (HCT-2116, MGC-803 and KYSE450) was greater than that of the positive controls (SAHA, TAE226). Finally, **30** showed better efficacy in in vivo studies (mice xenograft MGC-803 tumor model), where it inhibited tumor growth more than SAHA, TAE-226, as well as their combination [166].

All previously mentioned dual HDAC6 inhibitors are listed in Table 1.

The most important interactions between dual HDAC6 inhibitors and both targets are presented in Table 2.

## 4. Present and Future Perspectives in Development of Dual/Multi-Target HDA6 Inhibitors

There are already several multi-target HDAC inhibitors in clinical trials (see Table 3) that have inhibitory activity against HDAC6, but none of them is selective for HDAC6 among the other HDACs.

This suggests that there is great potential for further development of dual or multi-target HDAC6 inhibitors with selectivity for HDAC6 among the other HDAC isoforms. Even if the numerous dual HDAC6 inhibitors are discovered, there is still much potential for the future in this field of research.

SIRT2 has become a potential molecular target for the treatment of several diseases such as neurodegenerative diseases, cancer and metabolic syndrome [174]. SIRT2 has been shown to be upregulated in some tumors such as leukemia, hepatocellular carcinoma, gastric carcinoma and melanoma, while it is downregulated in ovarian cancer, prostate cancer and glioma [175]. Previous data showed the great potential of combining HDAC6 and SIRT2 inhibitors in the treatment of various tumors. In their recent research, Moon Hee Yang and coauthors demonstrated the association between the activity of HDAC6 and SIRT2 and the oncogenic activity of mutant K-RAS, which is highly prevalent in high-mortality cancers. The high degree of acetylation of lysine 104 in mutant K-RAS prevents its complete activation and oncogenic activity [176]. Moon Hee Yang and coauthors showed that HDAC6 and SIRT2 are the key enzymes controlling the acetylation of K-RAS. Thus, simultaneous inhibition of these enzymes prevents the deacetylation of lysine 104, leading to a decrease in the oncogenic activity of K-RAS [176]. Moreover, North et al. showed that HDAC6 and SIRT2 regulate the balance of α-tubulin acetylation, which correlates with cancer migration and invasion [177]. Recently, Sinatra L. and coauthors presented a first-in-class dual HDAC6/SIRT2 inhibitor. It showed great potency against both enzymes—IC_50_ = 0.32 µM for SIRT2 and IC_50_ = 0.043 µM for HDAC6, respectively. Moreover, this has better antiproliferative effect compared to single or combination therapy (selective SIRT2 + selective HDAC6 inhibitor) in the treatment of W1 ovarian cancer cells [178].

Glutamine is a very important source of energy for rapidly dividing cells and, together with its metabolites such as glutamate and α-ketoglutarate, plays an important role in the biosynthesis of macromolecules (nucleic acids, proteins and lipids) [179]. Some tumors may develop glutamine addiction. Besides its role in metabolism, glutamine may also be involved in tumorigenesis in other ways, such as its influence on transcription factor function (STAT3) and its involvement in the mTORC1 pathway (mammalian target of rapamycin complex 1 pathway) [180]. Two isoforms of glutaminases are known: renal type glutaminase (GLS1) and liver-type glutaminase (GLS2). They play opposite roles in tumorigenesis: GLS1 stimulates tumor growth, while GLS2 plays a tumor suppressive role [179,180]. Quin Q. and coauthors reported that the combination of an HDAC inhibitor and a GLS1 inhibitor destroys leukemia stem cells through a synergistic mechanism [181]. Previous studies have shown that the inhibition of HDAC6 increases GLS1 levels, which may be important for tumor progression [181]. Therefore, the use of this combination could be beneficial for patient therapy. Dual HDAC6/GLS1 inhibitors have not yet been discovered, which represents an opportunity for further studies.

## 5. Conclusions

Multi-targeting ligands have ushered in a new era in tumor therapy. This therapy approach could overcome the major drawbacks of single-target therapy and drug combinations, such as drug resistance, drug–drug interactions, adverse effects and toxicity, difficulties in compound pharmacokinetic profiling and more. In this review, the significance of dual inhibitors targeting one of the most important epigenetic regulators—histone deacetylases 6 (HDAC6)—were presented. Considering the role of HDAC6 in maintaining the balance of acetylation of non-histone and histone proteins, the localization of HDAC6, the specific structure of catalytic domain 2 and the data showing that knockout mice survive well without HDAC6, this isoform was selected before the others. Some of the single inhibitors selective for HDAC6 are in clinical trials for the treatment of solid and hematologic malignancies, whereas the five pan-HDAC inhibitors are approved for clinical use. A poor safety profile and drug resistance are the main disadvantages of pan-HDAC inhibitors, whereas HDAC6 inhibitors require high concentrations to exhibit anticancer properties that may lead to off-target effects. Therefore, the dual-target approach and selectivity for one HDAC isoform, primarily HDAC6, among the others have been the focus of recent research efforts to overcome the above problems. Numerous dual HDAC6 inhibitors have already been reported and discussed, such as dual BRD4/HDAC6, PI3K/HDAC6, HSP90/HDAC6, AR /HDAC6, tubulin/HDAC6, mTOR/HDAC6, HDAC1/6, HDAC3/6, HDAC8/6, FAK1/HDAC6, PAK1/HDAC6 inhibitors, and many of them have shown great efficacy in in vitro studies, with some of them displaying great potential in in vivo studies compared to single-targeted therapy or drug combinations.

However, the dual-target approach faces several challenges. The most important issue is how to select the optimal target to combine with HDAC6. In addition to target selection, it is very difficult to combine the pharmacophores of the two targets and design a molecule with high selectivity for the selected targets. Besides the selectivity of dual inhibitors, the balance of the activity of the ligand toward two targets can also be a major problem that requires lead optimization. Moreover, the pharmacokinetic properties of novel dual HDAC6 inhibitors should be considered taking into account the poor pharmacokinetic profile of already known HDAC inhibitors, which can be associated with a hydroxamic acid commonly used as the ZBG. Clearly, the optimal development of a dual HDAC6 inhibitor may be challenging. Despite these challenges, the development of dual HDAC6 inhibitors could be important for further tumor treatment and overcoming the problems of the currently approved therapy with pan-HDAC inhibitors.

## Figures and Tables

**Figure 1 pharmaceutics-15-02581-f001:**
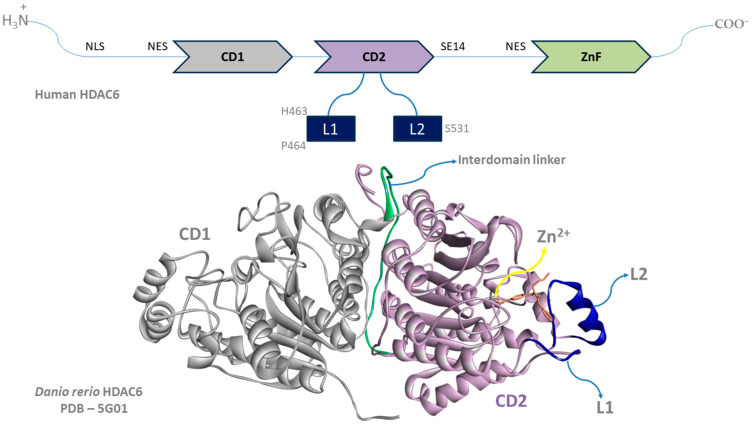
The structure of human HDAC6 (above) and danio rerio HDAC6 (below).

**Figure 2 pharmaceutics-15-02581-f002:**
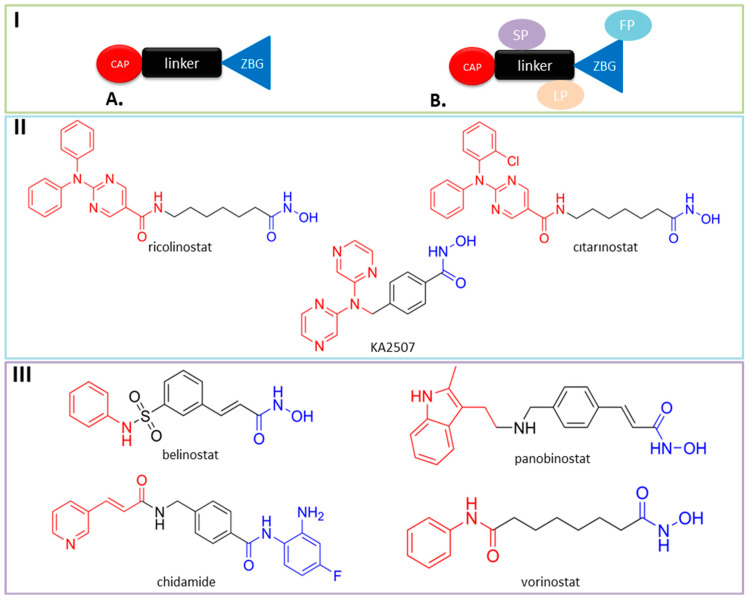
(**IA**) Classical pharmacophore model of HDAC inhibitors; (**IB**) extended pharmacophore model of HDAC inhibitors; (**II**) structures of selective HDAC6 inhibitors that are in clinical trials; (**III**) structures of registered pan-HDAC inhibitors.

**Figure 3 pharmaceutics-15-02581-f003:**
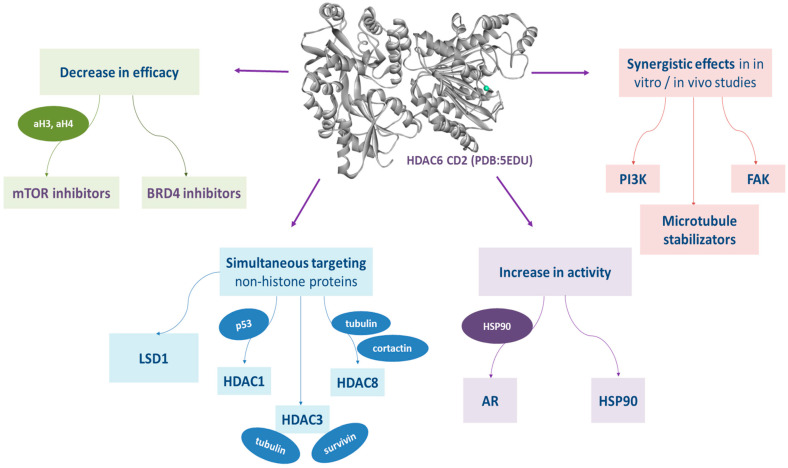
Schematic representation of molecular associations and signaling pathways of HDAC6 and other targets.

**Figure 4 pharmaceutics-15-02581-f004:**
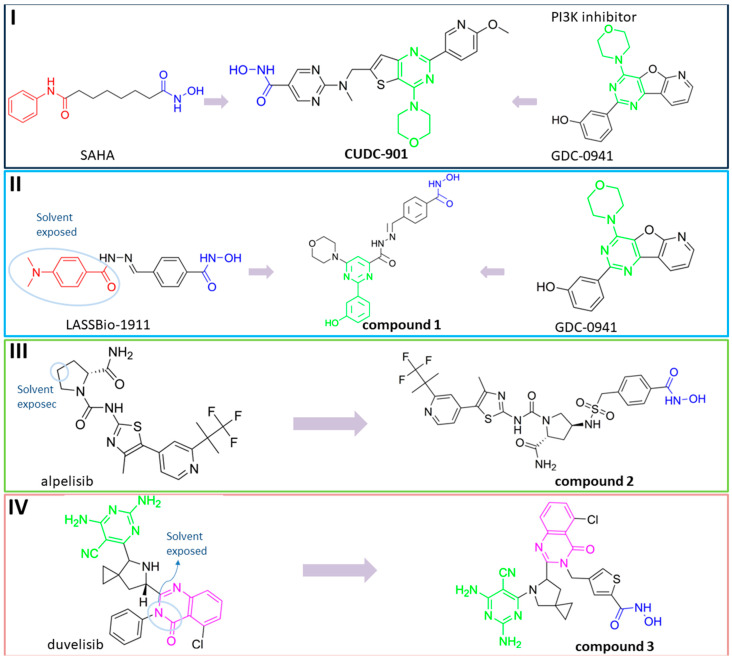
Development of dual HDAC6/PI3K inhibitors; (**I**,**II**)—fusion of pharmacophores of PI3K inhibitors and HDAC6 inhibitors in order to develop HDAC6/PI3K dual inhibitors (CUDC-901 and **1**); (**III**)—Modification of alpelisib structure in order to develop HDAC6/ PI3Kα dual inhibitor—**2**; (**IV**)—Modification of duvelisib structure in order to develop HDAC6 PI3Kδ dual inhibitor—**3**; The zinc-binding groups are highlighted in **blue**; the PI3K inhibitors’ key structure components are highlighted in **green** and **pink**; the CAP group of HDAC inhibitors is highlighted in **red;** and the solvent-exposed regions are indicated by **blue** circles.

**Figure 5 pharmaceutics-15-02581-f005:**
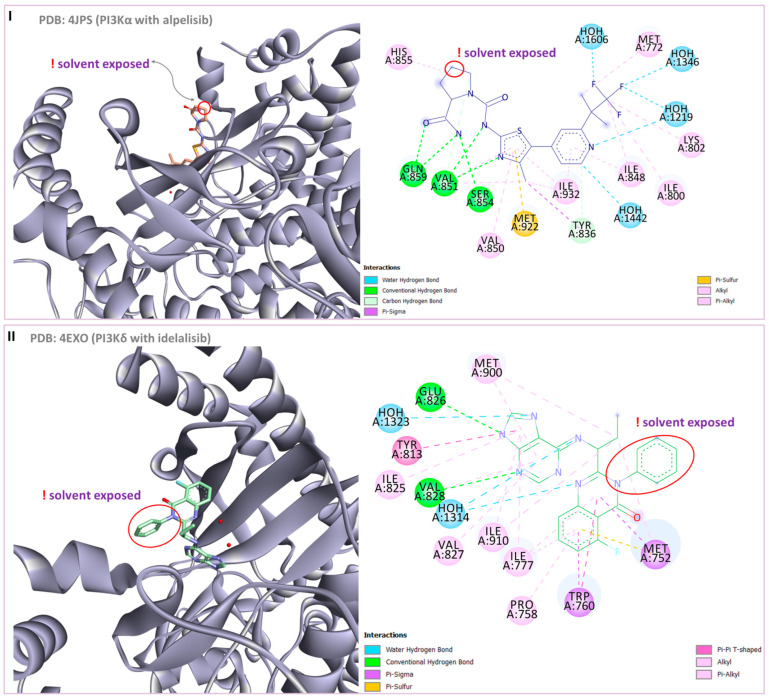
(**I**) Basis for rational design of **2**: position of alpelisib inside the binding pocket of PI3Kα (**left**) and 2D diagram of the most important alpelisib–PI3Kα interactions (**right**); solvent-exposed region is indicated by red circles. (**II**) Basis for rational design of **3**: position of idelalisib inside the binding pocket of PI3Kδ (**left**) and 2D diagram of the most important idelalisib–PI3Kα interactions (**right**); the solvent-exposed region is indicated by red circles.

**Figure 6 pharmaceutics-15-02581-f006:**
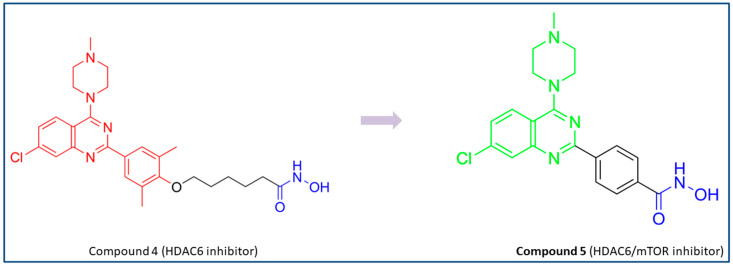
Development of dual HDAC6/mTOR inhibitor. The zinc-binding group is highlighted in **blue**; the hinge domain is highlighted in **green**; and the CAP group is highlighted in **red**.

**Figure 7 pharmaceutics-15-02581-f007:**
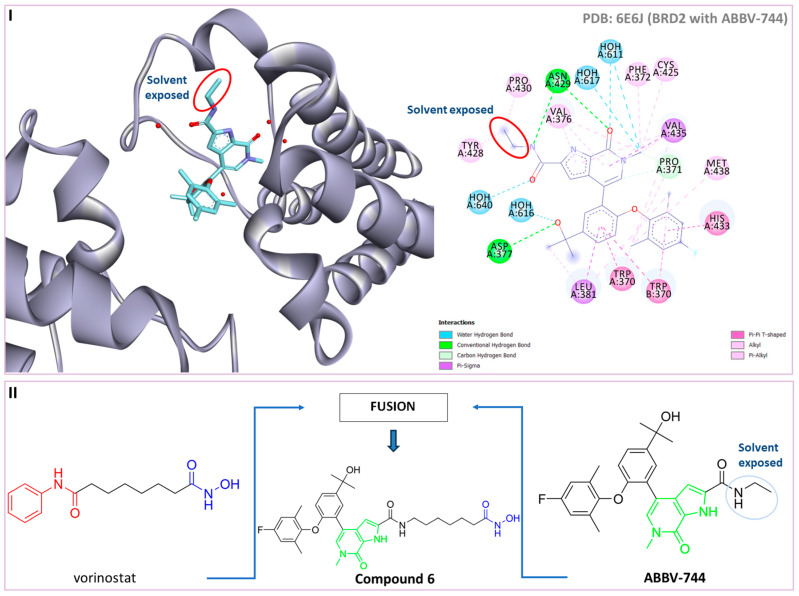
(**I**) Basis for rational design of dual HDAC6/BRD4 inhibitor: position of ABBV-744 inside the binding pocket of bromodomain 2 (BD2) (left) and 2D diagram of the most important ABBV744–BD2 interactions (right); the solvent-exposed region is indicated by red circles; (**II**) Development of dual HDAC6/BRD4 inhibitor: fusion of pharmacophores of BRD4 inhibitor and HDAC6 inhibitor. The zinc-binding group is highlighted in **blue**; the BRD4 inhibitor’s key structure components are highlighted in **green**; the CAP group of HDAC inhibitor is highlighted in **red**; and the solvent-exposed part of ABBV-744 is indicated by a **blue** circle.

**Figure 8 pharmaceutics-15-02581-f008:**
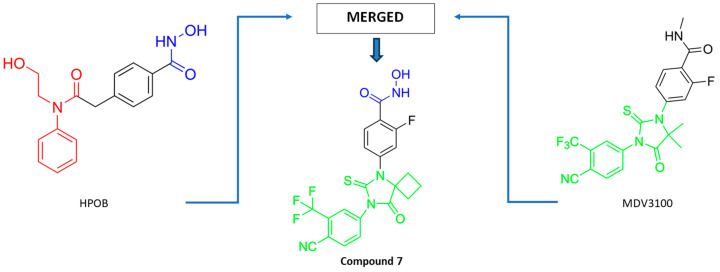
Development of dual HDAC6/AR inhibitor: merger of pharmacophores of AR inhibitor and HDAC6 inhibitor. The zinc-binding group is highlighted in **blue**; the CAP group is highlighted in **red**, and the pharmacophore group of AR inhibitor is highlighted **green**.

**Figure 9 pharmaceutics-15-02581-f009:**
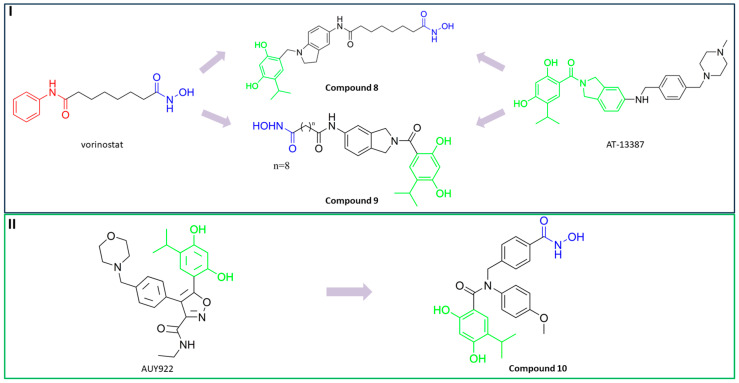
Development of dual HDAC6/HSP90 inhibitors. (**I**) Fusion of pharmacophores of HSP90 inhibitor and HDAC inhibitor. The zinc-binding group is highlighted in **blue**; the CAP group is highlighted in **red**; and the HSP90 binding domain is highlighted in **green**. (**II**) Modification of AUY922 structure in order to develop HDAC6/HSP90 dual inhibitor—**10**.

**Figure 10 pharmaceutics-15-02581-f010:**
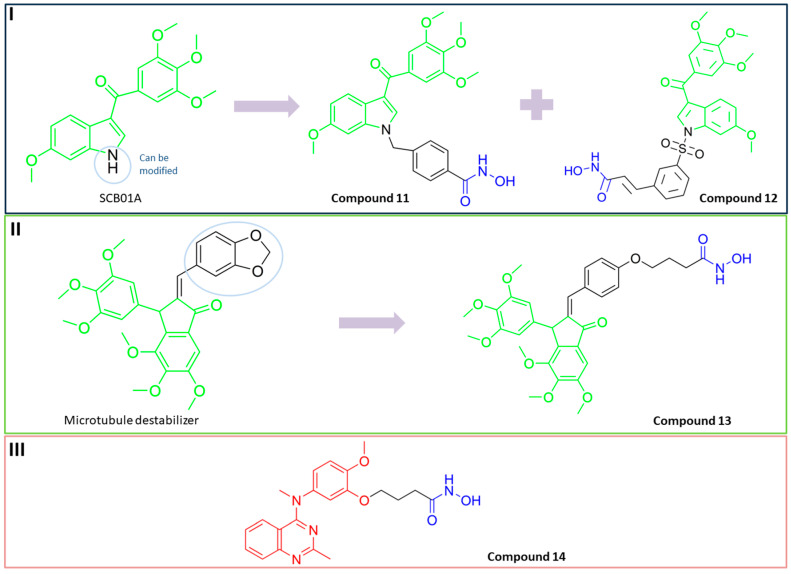
Development of dual HDAC6/tubulin inhibitors: (**I**)—Modification of SCB01A structure in order to develop HDAC6/tubulin dual inhibitors—**11** and **12**; (**II**)—Modification the structure of already known microtubule destabilizer in order to design **13**; (**III**)—Structure of HDAC6/tubulin dual inhibitor (**14**)The zinc-binding groups are highlighted in **blue**; the tubulin inhibitors’ key structure components are highlighted in **green**; the CAP group of HDAC inhibitors is highlighted in **red**; and the solvent-exposed regions are indicated with **blue** circles.

**Figure 11 pharmaceutics-15-02581-f011:**
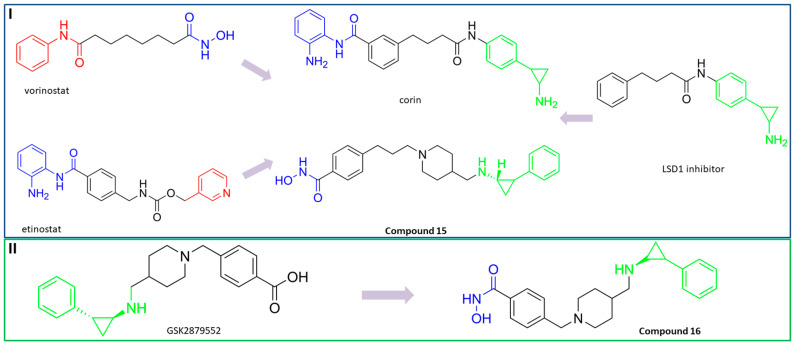
Development of dual HDAC6/LSD1 inhibitors: (**I**)—fusion of pharmacophores of LSD1 inhibitor and HDAC inhibitors in order to develop **corin** and **15**; (**II**)—Modification of GSK2879552 structure (already known LSD1 inhibitor) in order to develop HDAC6/LSD1 dual inhibitor—**16**. The zinc-binding groups are highlighted in **blue**; the CAP groups are highlighted in **red**; and the critical part of the structure important for LSD1 inhibition is highlighted in **green**.

**Figure 12 pharmaceutics-15-02581-f012:**
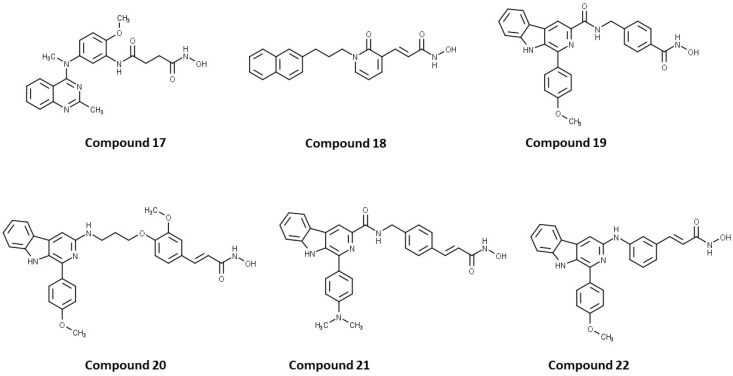
The most potent HDAC6/1 inhibitors from each series.

**Figure 13 pharmaceutics-15-02581-f013:**
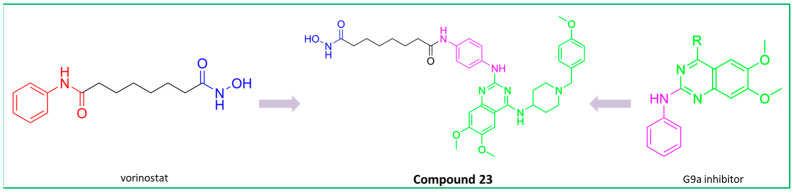
Development of dual HDAC6/HDAC3 inhibitors: fusion of pharmacophores of G9a inhibitor and HDAC inhibitors. The zinc-binding groups are highlighted in **blue;** the CAP group is highlighted in **red**; and the pharmacophore of G9a inhibitor is highlighted in **green** and **pink**.

**Figure 14 pharmaceutics-15-02581-f014:**
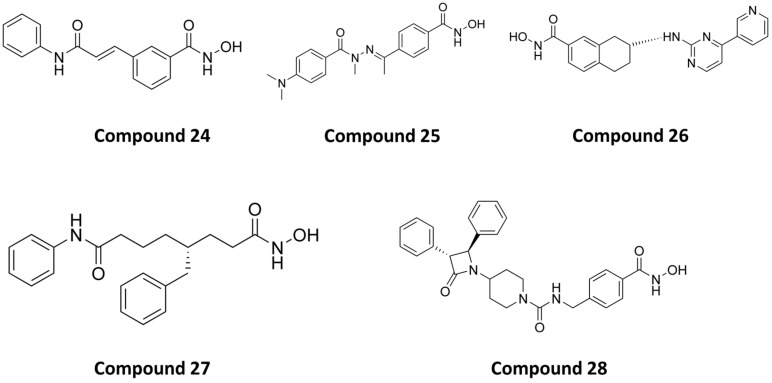
The most potent HDAC6/8 inhibitors from each series.

**Figure 15 pharmaceutics-15-02581-f015:**
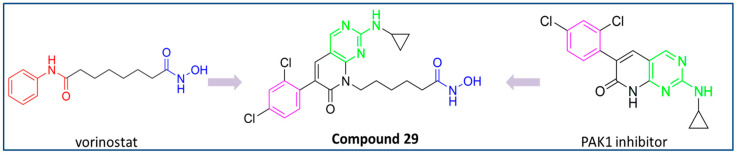
Development of dual HDAC6/PAK1 inhibitors: fusion of pharmacophores of PAK1 inhibitor and HDAC6 inhibitors. The zinc-binding groups are highlighted in **blue**; the CAP group of HDAC inhibitor is highlighted in **red**, the hinge domain of PAK1 inhibitor is highlighted in **green;** and the structure features that interact with hydrophobic pocket are highlighted in **purple**.

**Figure 16 pharmaceutics-15-02581-f016:**
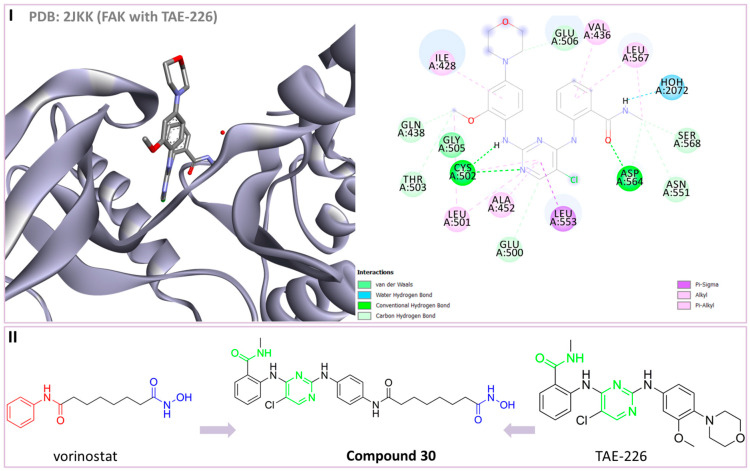
(**I**) Basis for rational design of dual HDAC6/FAK inhibitor: position of TAE-226 inside the binding pocket of FAK (**left**) and 2D diagram of the most important TAE-226–FAK interactions (**right**); the solvent-exposed region is highlighted in red. (**II**) Development of dual HDAC6/FAK inhibitor: fusion of pharmacophores of FAK inhibitor and HDAC6 inhibitors. The zinc-binding groups are highlighted in **blue**; the CAP group of HDAC inhibitors is highlighted in **red;** and the key structural elements important for FAK inhibition are highlighted in **green**.

**Table 1 pharmaceutics-15-02581-t001:** Dual HDAC6 inhibitors.

Structure/PubChem ID	Targets	IC_50_ Values	In Vivo	Ref.
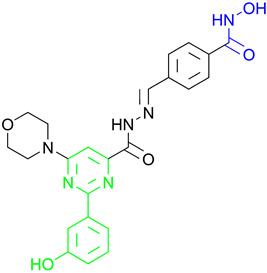 PubChem CID: 168289467 (**1**)	HDAC6/PI3Kα	15.3 nM (HDAC6) 46.3 nM (PI3Kα)	No data	[71]
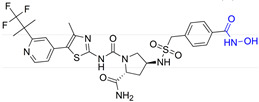 **2**	HDAC6/PI3Kα	26 nM (HDAC6) 2.9 nM (PI3Kα)	No data	[72]
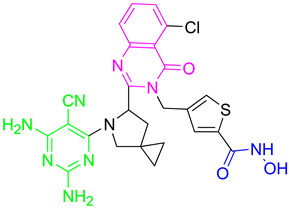 PubChem CID: 168298545 (**3**)	HDAC6/PI3Kδ	2.3 nM (PI3Kδ) 13 nM (HDAC6)	No data	[70]
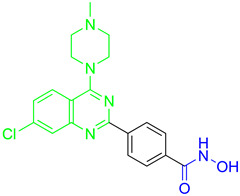 PubChem CID: 163322242 (**5**)	HDAC6/mTOR	56 nM (HDAC6) 133.7 nM (mTOR)	No data	[73]
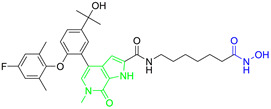 PubChem CID: 155431228 (**6**)	HDAC6/ BRD4	17.2 nM (HDAC6) 1.2 µM (BRD4)	No data	[82]
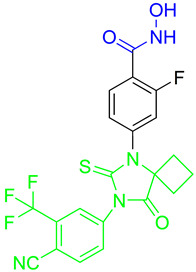 PubChem CID: 146434737 (**7**)	HDAC6/AR	HDAC6 (0.98 µM) AR (0.63 µM)	Stronger antitumor activity compared to positive control (MDV3100) in CRPC xenograft model	[86]
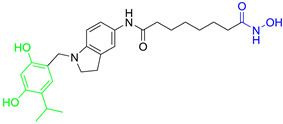 PubChem CID: 145985243 (**8**)	HDAC6/HSP90	1.15 nM (HDAC6) 46.3 nM (HSP90)	No data	[91]
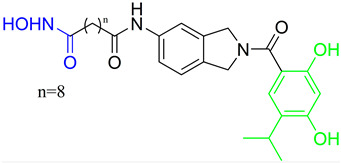 PubChem CID: 145984665 (**9**)	HDAC6/HSP90	4.32 nM (HDAC6) 46.8 nM (HSP90)	TGI = 44.8% (50 mg/kg of **9**); TGI = 69.7% (100 mg/kg of **9**); TGI = 72.3% (50 mg/kg of **9**) in combination with afatinib (20 mg/kg); human NSCLC H1975 xenograft model	[89]
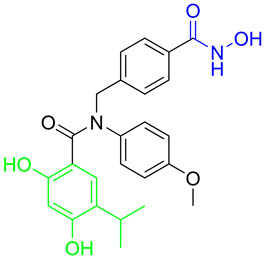 PubChem CID: 167505967 (**10**)	HDAC6/HSP90	4.56 nM (HDAC6) 52 nM (HSP90)	TGI = 83.9% (50 mg/kg of **10**) in combination with anti-PD-1 (programmed cell death protein 1) antibodies (200 µg); BALB/c mice-bearing CT26 tumor (colorectal cancer)	[92]
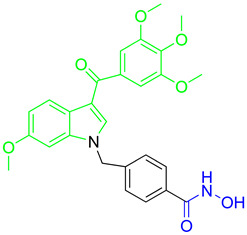 PubChem CID: 137654946 (**11**)	HDAC6/ tubulin	64.5 nM (HDAC6)	TGI = 24.8% (100 mg/kg) and TGI = 68.5% (200 mg/kg)—human prostate xenograft nude mouse model (PC3); TGI = 35.5% (50 mg/kg) and TGI = 58.2% (100 mg/kg)—multiple myeloma xenograft model (RPMI-8226)	[95]
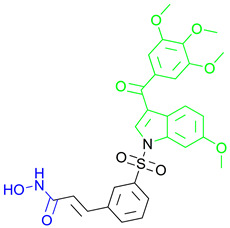 PubChem CID: 137645287 (**12**)	HDAC6/ tubulin	275.35 nM (HDAC6)	TGI = 40.9% human leukemia mouse xenograft model (HL-60) TGI = 31.1%—PC3 grafted mice model	[96]
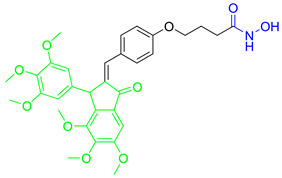 **13**	HDAC6/ tubulin	residual HDAC6 activity at 20 µM was 3%	No data	[98]
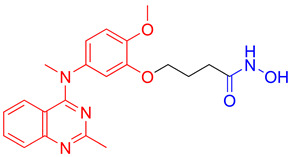 PubChem CID: 122550211 (**14**)	HDAC6/ tubulin	17 nM (HDAC6)	TGI = 66.05% (50 mg/kg of **14**)—HCT116 model; TGI = 77.39% (25 mg/kg of **14**)—A2780s model; TGI = 65.65% (50 mg/kg of **14**)—MCF-7 model	[99]
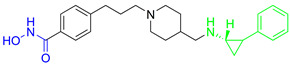 PubChem CID: 132138171 (**15**)	HDAC6/LSD1	48 nM (HDAC6) 6 nM (LSD1)	TGI = 67% (25 mg/kg of **15**)—MM 1.S xenograft model	[112]
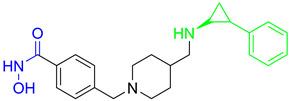 PubChem CID: 132118589 (**16**)	HDAC6/LSD1	0.11 µM (HDAC6) 0.54 µM (LSD1)	No data	[113]
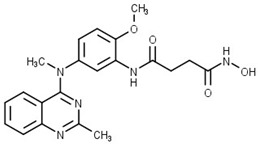 **17**	HDAC6/HDAC1	16.15 nM (HDAC6) 31.1 nM (HDAC1)	No data	[127]
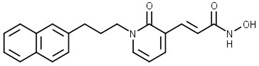 PubChem CID: 49847150 (**18**)	HDAC6/HDAC1	2.46 nM (HDAC6) 19.4 nM (HDAC1)	No data	[128]
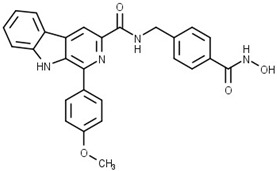 **19**	HDAC6/HDAC1	2.6 nM (HDAC6) 4.1 nM (HDAC1)	Treatment with **19** inhibited growth of hepatoma tumor that was comparable to positive control SAHA at the same dose	[129]
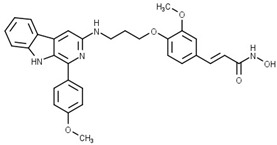 PubChem CID: 155515836 (**20**)	HDAC6/HDAC1	13 nM (HDAC6) 27 nM (HDAC1)	No data	[130]
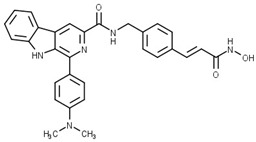 **21**	HDAC6/HDAC1	7.6 nM (HDAC6) 29 nM (HDAC1)	TGI = 65% (70 µmol/kg of **21**)—Bel7402/5-FU xenograft tumor model	[131]
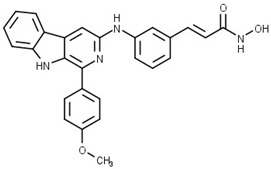 **22**	HDAC6/HDAC1	3.1 nM (HDAC6) 1.3 nM (HDAC1)	No data	[132]
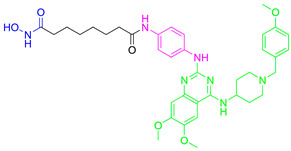 PubChem CID: 155556138 (**23**)	HDAC6/HDAC3	34 nM (HDAC6) 2.6 nM (HDAC3)	No data	[143]
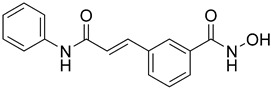 PubChem CID: 71681069 (**24**)	HDAC6/HDAC8	21 nM (HDAC6) 37 nM (HDAC8)	No data	[156]
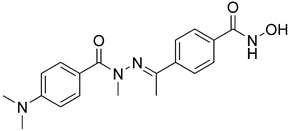 PubChem CID: 154487896 (**25**)	HDAC6/HDAC8	97 nM (HDAC6) 54 nM (HDAC8)	No data	[157]
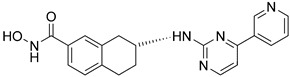 PubChem CID: 92045024 (**26**)	HDAC6/HDAC8	50 nM (HDAC6) 80 nM (HDAC8)	No data	[158]
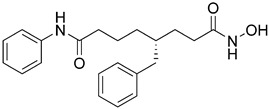 PubChem CID: 134283800 (**27**)	HDAC6/HDAC8	48 nM (HDAC6) 27 nM (HDAC8)	No data	[159]
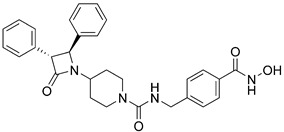 PubChem CID: 168282542 (**28**)	HDAC6/HDAC8	21 nM (HDAC6) 42 nM (HDAC8)	Chemical toxicity was evaluated by zebrafish model. **28** was well tolerated up to 75 µM, which confirmed its safety profile	[162]
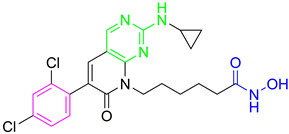 PubChem CID: 168510343 (**29**)	HDAC6/PAK1	38.23 nM (HDAC6) 13.62 nM (PAK1)	Great potential of **29** was confirmed by in vivo studies—in the TNBC xenograft zebrafish and mouse model	[164]
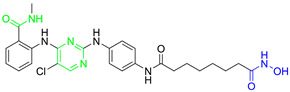 **30**	HDAC /FAK	16 nM (HDAC6) 132 nM (FAK)	TGI = 53.5% (20 mg/kg of **30**)—gastric cancer cells MGC-803 xenograft model	[166]

**Table 2 pharmaceutics-15-02581-t002:** The key interactions between dual HDAC6 inhibitors and targets obtained by molecular docking studies.

Cd	Interactions with HDAC6	Interactions with Another Target	Ref.
1	/	PI3Kα: hydrogen bonds with Val851, tyrosine 836, aspartate 810.	[71]
2	PDB:5EDU—hydroxamic acid coordinates the zinc ion; hydrogen bonds with His610, His611 and Tyr782; π-π interactions are established with Phe620 and Phe680.	PI3Kα (PDB:4JPS): key hydrogen bonds with Val851, Ser854 and Gln859; nitrogen atom of pyridine forms hydrogen bonds with Lys802 and Asp810; hydrogen bond is established between fluorine atoms and Lys802	[72]
3	Hydroxamic acid coordinates zinc; this complex is additionally stabilized by hydrogen bond with Tyr745; π-π interactions are established between the linker and Phe583, His614, Phe643 residues; π-π interactions are established between CAP group and His463 and Pro464.	PI3Kδ: hydrogen bonds with hinge region (Glu826 and Val828); π-π interactions are established with Met752 and Trp760.	[70]
5	Hydroxamic acid chelates the zinc ion, and this complex is additionally stabilized by hydrogen bond with the His573; π-π interactions are formed with Phe583 and Phe643.	mTOR: salt bridge interactions are observed with Asp2195, Asp2357 and Glu2190 residues and hydrogen bonds with Val2240 and Trp2239 residues.	[73]
7	Compound **7** interacts with HDAC6 in a manner similar to that of already known HDAC6 inhibitor—HOBP.	AR: compound **7** inserts in AR in a manner similar to that of bicalutamide.	[86]
9	Hydrogen bonds are described between CAP group and Ser546, Phe566 and Ile569; hydrophobic interactions are established with Phe620; hydroxamic acid coordinates zinc; and this complex is stabilized by hydrogen bonds with His610 and Gly619.	HSP90: hydrogen bonds are established between Asn51, Lys58 Asp93, Gly108, Thr184 and 2,4-dihydroxy-5-isopropybenzoyl moiety; hydrophobic interactions are established with Ala55, Met98, Thr109.	[89]
12	PDB:5EDU—N-hydroxyformamide moiety forms a complex with the zinc ion; hydrogen bonds are established with residues His610 and His611; hydrophobic interactions are established with residues Ser568, Gly619, Phe620, His651 and Phe680.	Colchicine binding site of tubulin (PDB:4O2B): hydrophobic interactions are formed with Met259, Ala316, Ile318 and Ile378; hydrogen bonds are established with Ser178 and Asp329 residues.	[96]
17	Π-cation interaction is observed between the quinazoline group and the phenyl group of Tyr1022; hydrophobic interactions are established with Asp1044 and Tyr1055 residues; hydroxamic acid forms a complex with the zinc ion.	HDAC1: hydrophobic interactions are formed with Lys331, Arg270 and Arg306 residues; hydroxamic acid forms a complex with the zinc ion.	[127]
25	Docking procedure is performed with a homology model built from HDAC7 (PDB ID: 1C0Z).	HDAC8 (PDB ID:1VKG)—a complex with the zinc ion is established as well as hydrogen bonds with His142, His143 and Tyr306; hydrophobic interactions are formed with Phe152 and Phe208 residues.	[157]
26	Hydrophobic interactions are formed with Phe620 and Phe680; hydroxamic acid coordinates the zinc ion.	HDAC8 (PDB ID:1VKG): hydrophobic interactions are established with Pro35, Phe152 and Tyr306 of HDAC8	[158]
28	Hydroxamic acid coordinates the zinc ion and forms hydrogen bonds with Gly619 and Tyr782; hydrogen bonds are established with Phe680; and π-π interactions are established between benzyl linker and His651, Phe620 and Phe680.	HDAC8: in addition to coordinating the zinc ion, hydroxamic acid forms hydrogen bonds with Gly151 and Tyr306; π-π interactions are formed between benzyl linker and Phe152 and Phe208. Π-cation interaction is established between phenyl substituent of β-lactam moiety and Lys202	[162]
29	Hydroxamic acid group forms two hydrogen bonds with the His573 residue and generates a chelate product with the zinc ion; π-alkyl interactions are established with Pro464, Pro711 and Leu712 residues.	PAK1: two conserved hydrogen interactions are observed with the key Leu3347 residue from the kinase hinge; hydrophobic interactions are formed with Val284, Met344, Val342 and Lys299 residues; π-sulfur interaction is established between phenyl-moiety and Met344, and a π-cation interaction is formed within Lys299; hydrogen bonds are described between hydroxamic acid and Asp393 and Asn394 residues.	[164]

**Table 3 pharmaceutics-15-02581-t003:** Multi-targeting HDAC inhibitors in clinical trials.

Compound	Structure	Phase of Clinical Trials	HDAC (IC_50_ in nM)	Other Targets (IC_50_ in nM)	Clinical Use
*CUDC-101*	* 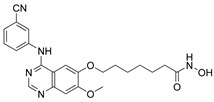 *	Phase 1 [167]	*HDAC1 (4.5); HDAC2 (12.6); HDAC3 (9.1); HDAC8 (79.8); HDAC4 (13.2); HDAC5 (11.4); HDAC6 (5.1); HDAC7 (373); HDAC9 (67.2); HDAC10 (26.1)* [168]	*EGFR (2.4); HER2 (15.7)* [168]	*Advanced solid tumors* [167]
*Tinostamustine (EDO-S101, NL-101)*	* 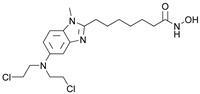 *	*Phase ½* [169,170]	*HDAC1 (9); HDAC2 (9); HDAC3 (25); HDAC8 (107); HDAC6 (6); HDAC10 (72)* [171]	*DNA alkylating agent* [171]	*Malignant hematological diseases and advanced solid tumors* [169,170]
*Fimepinostat (CUDC-907)*	* 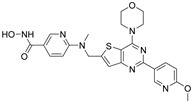 *	*Phase ½ (completed)* *Phase 1—active* [172,173]	*HDAC1 (1.7); HDAC2 (5.0); HDAC3 (1.8); HDAC8 (191); HDAC4 (409); HDAC5 (674); HDAC6 (27); HDAC7 (426); HDAC9 (554); HDAC10 (2.8);* *HDAC1 (5.4)* [69]	*PI3Kα (19); PI3Kβ (54); PI3Kδ (39); PI3Kγ (311)* [69]	*Advanced, relapsed and refractory solid tumors, CNS tumors, lymphoma* [172,173]

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
