# Peer review of "Targeting Histone Deacetylases 6 in Dual-Target Therapy of Cancer"

_pharmaceutics, 2023, doi:10.3390/pharmaceutics15112581_

Round 1
Reviewer 1 Report
Comments and Suggestions for Authors
This manuscript is a review of dual inhibitors targeting the epigenetic enzyme histone deacetylase 6 (HDAC6) and other histone deacetylases for targeted cancer therapies. The paper is very well-written, and the subject matter organized in a nice manner. Many compound structures are included as figures, and the review would be an asset to anyone working in strategic design of histone deacetylases. Only a couple of minor spelling errors were noted.
The manuscript is a review of the literature on advances in cancer treatment with HDAC6 inhibitors in combination with 10 different targets. The review is particularly relevant to area of dual-targeting approach to cancer therapy with HDAC6 inhibitors and is important for addressing challenges of the currently approved therapy with pan-HDAC inhibitors.
The review provides a comprehensive compilation of pharmacophores of ligands targeting HDAC6 in approaches to screen against known targets.
The paper is a review of the published literature and does not include any original research; therefore, experimental methodology, including control groups, is not relevant to the review.
The chemical structures of the many pharmacophores of HDAC6 inhibitors are nicely presented in a well-organized manner that would serve as a resource for researchers in drug discovery and development of HDAC6 inhibitors in multi-target cancer therapy.
Author Response
Dear reviewer,
Thank you for your comments regarding this manuscript.
Kind regards

Reviewer 2 Report
Comments and Suggestions for Authors
Histone deacetylases (HDACs) are the major regulators of the balance of acetylation of histone and non-histone proteins. In contrast to the other HDAC isoforms, HDAC6 is mainly involved in maintaining the acetylation balance of many non-histone proteins. Therefore, overexpression of HDAC6 has been associated with tumorigenesis, invasion, migration, survival, apoptosis, and growth of various malignancies. As a result, HDAC6 is considered a promising target for cancer treatment. However, none of the selective HDAC6 inhibitors are in clinical use, mainly because of drug resistance, which is a typical problem of single-targeted therapy. Herein, the authors summarized the advances in tumor treatment with dual HDAC6 inhibitors, which provide a new direction for solve the problem of drug discovery against HDAC6. But I have several following concerns:
1. This manuscript mainly reviews the application of HDAC6 in dual-target therapy. It is suggested that the title be modified to "Targeting Histone deacetylases 6 in dual-target therapy of cancer".
2. Abbreviations should be defined when they first appear in the text. Such as "BRAF", "MEK"...
3. It is suggested that the authors compile the information of each dual inhibitors in Section 3 of the article, including name, target, IC50 value, efficacy in cellulo or in vivo, and references in a Table.
4. In Line 678, "EC50" should be "EC50"
5. Table 1 should use a standard three-line table and should not span pages.
6. Please unify the format of references in the article, including the author's name, the case of words in the title of the article, the writing of the name of the journal, and the page number.
Comments on the Quality of English LanguageMinor editing of English language required.
Author Response
Dear reviewer,
Thank you for your comments which help us to improve manuscript. Please find below answers to all comments:
- This manuscript mainly reviews the application of HDAC6 in dual-target therapy. It is suggested that the title be modified to "Targeting Histone deacetylases 6 in dual-target therapy of cancer".
Thank you very much for your comment. We are in agreement with you. Therefore, we have changed the title of the manuscript. The title of the manuscript is now "Targeting Histone deacetylases 6 in dual-target therapy of cancer".
- Abbreviations should be defined when they first appear in the text. Such as "BRAF", "MEK".
Thank you for your comment. Abbreviations are now defined when they first appear in the text. For example, BRAF (B-Raf serine-threonine kinase) is defined on page 5, line 163; MEK (mitogen-activated extracellular signal-regulated kinase) is defined on page 5, lines 163 and 164; ROS1 (receptor tyrosine kinase) is defined on page 5, line 174; p300/CBP(E1A binding protein p300/CREB-binding protein), is defined on page 19, line 689, etc.
- It is suggested that the authors compile the information of each dual inhibitors in Section 3 of the article, including name, target, IC50 value, efficacy in cellulo or in vivo, and references in a Table.
Thank you very much for your comment. We have compiled the information on each dual HDAC6 inhibitor and you can find it in Table 1.
- In Line 678, "EC50" should be "EC50"
Thank you for your comment. EC50 is modified into EC50 in lines 757 and 758.
- Table 1 should use a standard three-line table and should not span pages.
Thank you for your comment. All tables in the manuscript (Table 1 – pages 23-29; Table 2 – pages 29 and 30; Table 3 – page 31) are in form of three-line table and they do not span pages.
- Please unify the format of references in the article, including the author's name, the case of words in the title of the article, the writing of the name of the journal, and the page number.
Thank you for your comment. All references have now been unified through the use of Zotero (Multidisciplinary Digital Publishing Institute style).
Kind regards,

Reviewer 3 Report
Comments and Suggestions for Authors
Authors of the presented manuscript provided evidence and literature cited studies for tumour treatment with multi-target Histone deacetylase (HDAC) inhibitors. This review article is comprehensive and considered valuable within its field of developing multi-target therapeutics for combating cancer. Suggestions are addressed as following;
1. Authors are advised to provide a schematic representation for the biological and molecular role of HDACs as well as its molecular associations and signaling pathways with the different biotargets (i.e. PI3K, mTOR, etc) being mentioned within the provided manuscript.
2. Authors should provide descriptive data regarding the HDAC topology and key secondary structures, motifs, and binding site, while being augmented with a 3D representation of them. Additionally, mutations and SNP variants of the HDAC target protein should be also comprehensively addressed.
3. Figures of dual HDAC inhibitors should be improved. Heteroatoms are in small sizes and difficult to be identified. Authors are advised to adopt the “ACS document object setting” for better compound representations or as seen with the compounds at Figure 9.
4. In Figure 5, it seems that the multi-target drug design approach is “Merged” rather than what the authors stated as “Fusion”, since lots of HDAC pharmacophoric features are missing.
5. In Table 1, structures of the reported dual HDAC inhibitors are truncated as the column size seems to be inappropriate.
6. Authors are advised to present graphical representation for compound-target complexes being either deposited in PDB database or even replicated through docking approaches. This would provide better tracking for the reported compound’s binding interaction with the key target residues and their successful translation into relevant biological significance.
Comments on the Quality of English LanguageMinor editing of English language required
Author Response
Dear reviewer,
Thank you for your comments which help us improve our manuscript. In the next sections you will find answers to your concerns:
- Authors are advised to provide a schematic representation for the biological and molecular role of HDACs as well as its molecular associations and signaling pathways with the different biotargets (i.e. PI3K, mTOR, etc) being mentioned within the provided manuscript.
Thank you for your comment. A schematic representation of the molecular relationships of HDAC6 and other targets can now be found in the manuscript (Figure 3).
- Authors should provide descriptive data regarding the HDAC topology and key secondary structures, motifs, and binding site, while being augmented with a 3D representation of them. Additionally, mutations and SNP variants of the HDAC target protein should be also comprehensively addressed.
Thank you very much for your comment. The topology, secondary structures, and binding site of HDAC6 are now descrbied in the Introduction section (lines 54-73, page 2) and supplemented by a 3D representation (Figure 1). You will find the part of this section below:
“Besides the distinctions in biological function and cellular localization of HDAC6 compared with other HDAC isoforms, HDAC6 also has a unique structure that can be exploited for the development of selective HDAC6 inhibitors (Figure 1). Unlike the other HDAC isoforms, HDAC6 contains two catalytic domains (CD1 and CD2), and its structure is characterised by the presence of a zinc finger domain with homology to ubiquitin-specific proteases that binds unanchored ubiquitin (ubiquitin-binding domain)[21,22]. In contrast to CD1, which is highly specific for substrates containing C-terminal acetyllysine residues (exo-acetyllysine peptide substrates), CD2 exhibits broader substrate specificity (exo- and endo-acetyllysine peptide substrates)[22]. Hai Y. and Christianson D. showed that a mutation in human CD2, but not in human CD1, reduces the catalytic activity of HDAC6 by more than 400-fold, indicating the importance of CD2 for the overall catalytic activity of HDAC6 [22]. Therefore, the binding modes and interactions with CD2 should be considered in the development of new HDAC6 inhibitors. Regarding the selectivity for the HDAC6 isoform among the other HDACs, previous studies have shown that the interactions between the CAP moiety of pan-HDAC inhibitors and H463 and P464 of the L1 loop are more important for binding to HDAC1, HDAC2, and HDAC3 than for HDAC6, which can be used to design selective HDAC6 inhibitors [22]. In addition, S531 (part of the L2 loop) has been identified as a crucial amino acid residue for the recognition of α-tubulin [21]. Therefore, this part of HDAC6 should also be considered in the development of novel HDAC6 inhibitors. “
- Figures of dual HDAC inhibitors should be improved. Heteroatoms are in small sizes and difficult to be identified. Authors are advised to adopt the “ACS document object setting” for better compound representations or as seen with the compounds at Figure 9.
Thank you for your comment. All the figures of the dual HDAC inhibitors are now improved by using “ACS document object setting”
- In Figure 5, it seems that the multi-target drug design approach is “Merged” rather than what the authors stated as “Fusion”, since lots of HDAC pharmacophoric features are missing.
Thank you very much for pointing out this error. It is corrected now and we used the phrase “merged” instead of “fusion" (line 401, page 12 and Figure 8)
- In Table 1, structures of the reported dual HDAC inhibitors are truncated as the column size seems to be inappropriate.
Thank you for your comment. We have changed the column size so that the structures of the reported dual HDAC inhibitors are not truncated (Table 3, page 31)
- Authors are advised to present graphical representation for compound-target complexes being either deposited in PDB database or even replicated through docking approaches. This would provide better tracking for the reported compound’s binding interaction with the key target residues and their successful translation into relevant biological significance.
Thank you very much for your advice. We have presented available drug-target complexes deposited in the PDB database, such as Alpelisib-PI3Kα (PDB:4JPS)–Figure 5-I, page 8; Idelalisib- PI3Kδ (PDB:4EXO)-Figure 5- II, page 8; ABBV744-BD2 (PDB:6E6J)–Figure 7- II, page 11 and TAE226-FAK (PDB:2JKK)-Figure 16-I, page 23. Information from the already known inhibitor-target complexes are used to design dual HDAC6 inhibitors with these targets. In addition, the key interactions between dual HDAC6 inhibitors and both targets, as determined by molecular docking studies, are listed in Table 2.
Kind regards
Reviewer 4 Report
Comments and Suggestions for Authors
Beljkas et al., presented the work titled as "Targeting Histone deacetylases 6 in multi-target therapy of cancer". HDACs are quite interesting and the work seems to be of worth. There are some comments which authors should implement:
1. Last two sentences of the abstract needs paraphrasing and in current form it lacks the endorsement of the work.
2. Please delete this: (List three to ten pertinent keywords specific to the article yet reasonably common within the subject discipline.)
3. Introduction heading is missing. I feel the authors must not directly jump to the point. Instead there must be summarized introduction then subheadings.
4. In section 1. Multi-target ... first paragraph is well written but I feel the need of 3-4 more relevant references in addition to the current number of references.
5. I will also recommend the authors to define and introduce the meaning and significance of dual inhibitors in summarized introduction section.
6. Please also insert one table for the PubChem id and references for the inhibitors.
7. This one line paragraph "HDAC6 controls microtubule dynamics ... " seems quite random if not then explain more and link the events.
8. Overall, I feel that the summarized introduction must also include some introduction about systems pharmacological approach which will link the overall story in more interesting way.
9. Some of the references, the authors should include in the main text are here:
https://translational-medicine.biomedcentral.com/articles/10.1186/1479-5876-10-S2-A18
https://www.annualreviews.org/doi/10.1146/annurev-pharmtox-011112-140248
https://www.ingentaconnect.com/content/ben/cchts/2020/00000023/00000008/art00009
https://www.mdpi.com/2073-4409/11/24/4121
Comments on the Quality of English Language
Minor issues need to be corrected.
Author Response
Dear reviewer,
Thank you for your comments, which help us improve our manuscript. In the next sections you will find answers to your concerns:
- Last two sentences of the abstract needs paraphrasing and in current form it lacks the endorsement of the work.
Thank you very much for your comment. The last two sentences have been changed and you can find this part of the summary below:
“However, none of the selective HDAC6 inhibitors are in clinical use, mainly because of the low efficacy and high concentrations used to show anticancer properties, which may lead to off-target effects. Therefore, HDAC6 inhibitors with dual- targeting capabilities represent a new trend in cancer treatment, aiming to overcome the above problems. In this review, we summarize the advances in tumor treatment with dual-targeting HDAC6 inhibitors”.
- Please delete this: (List three to ten pertinent keywords specific to the article yet reasonably common within the subject discipline.)
Thank you very much for your comment. We have deleted this part of abstract.
- Introduction heading is missing. I feel the authors must not directly jump to the point. Instead there must be summarized introduction then subheadings
Thank you very much for your comment. It helps us a lot to improve the manuscript. We have also added a Histone deacetylases 6 – a viable target for cancer section to the introduction heading to improve the quality of the manuscript and to link the story in a more meaningful way.
- In section 1. Multi-target ... first paragraph is well written but I feel the need of 3-4 more relevant references in addition to the current number of references.
Thank you for your comment. We have added other relevant references such as:
Han-Chung, W.; Chang, D.-K.; Chia-Ting, H. Targeted Therapy for Cancer. J. Cancer Mol. 2006, Lee, Y.T.; Tan, Y.J.; Oon, C.E. Molecular Targeted Therapy: Treating Cancer with Specificity. Eur. J. Pharmacol. 2018, 834, 188–196, doi:10.1016/j.ejphar.2018.07.034.
Gerber, D.E. Targeted Therapies: A New Generation of Cancer Treatments. Target. Ther. 2008, 77.
Mobashir, M.; Turunen, S.P.; Izhari, M.A.; Ashankyty, I.M.; Helleday, T.; Lehti, K. An Approach for Systems-Level Understanding of Prostate Cancer from High-Throughput Data Integration to Pathway Modeling and Simulation. Cells 2022, 11, 4121, doi:10.3390/cells11244121.
- I will also recommend the authors to define and introduce the meaning and significance of dual inhibitors in summarized introduction section
Thank you very much for your comment. We have introduced the meaning and significance of dual inhibitors in the summarized introductory section – (lines 105-137, pages 3 and 4). You will find this part of Introduction section bellow:
“Even if selective HDAC6 inhibitors have been developed, they have limited success in clinical trials as single-target therapy. The concerns have been raised about the use of HDAC6 inhibitors as single agents because high concentrations are used to show anticancer properties [33]. These high concentrations have been shown to impair selectivity for HDAC6 which may lead to off-target effects. On the other hand, adverse effects and off-target toxicities limited the clinical use of pan-HDAC inhibitors due to non-selectivity. Considering all the above reasons, there are two trends in the development of HDAC inhibitors:
- Increasing the selectivity toward one HDAC isoform (HDAC6) among others with the goal of reducing adverse effects
- Development of dual-targeting HDAC inhibitors in order to increase the efficacy and decrease the dose of HDAC6 inhibitors due to synergistic and additive effects.
Therefore, in this review, we will focus on dual inhibitors, all targeting the epigenetic enzyme histone deacetylase 6 (HDAC6) and one of the following targets, such as phosphati-dylinositol 3’-kinases (PI3K), mammalian target of rapamycin (mTOR), bro-mo-domain-containing proteins 4 (BRD4), androgen receptor (AR), heat shock protein 90 (HSP90), tubulin, lysine-specific demethlylase 1 (LSD1), p-21 activated kinases 1 (PAK1), focal adhesion kinase (FAK), histone deacetylases 1 (HDAC1), histone deacetylases 3 (HDAC3), and histone deacetylases 8 (HDAC8). The main reasons for the rational design (Figure 3) of dual inhibitors of HDAC6 and the one of the previously mentioned targets are:
- The synergistic effects of HDAC6 inhibitors in combination with PI3K, FAK inhibitors and microtubule stabilizators demonstrated in in vitro and/or in vivo studies [34–36]
- Decrease in efficacy of mTOR inhibitors and BRD4 inhibitors as single-target therapy due to overactivity of HDAC6[37,38];
- Increase in activity of AR and HSP90 due to overexpression of HDAC6[39,40];
- Simultaneous targeting of non-histone proteins by inhibiting HDAC1, HDAC3, HDAC8, LSD1 and HDAC6 may show synergistic effects on cancer cell lines[41–46].
All of the dual inhibitors presented in this review are selective for the HDAC6 isoform among the other histone deacetylases”.
- Please also insert one table for the PubChem id and references for the inhibitors.
Thank you for your comment. PubChem ID and references of dual HDAC6 inhibitors can now be found in Table 1. However, we could not find PubChem ID for some compounds, such as compounds 2, 13, 17, 19, 21, 22, and 30 but the references for all dual HDAC6 inhibitors are listed in Table 1.
- This one line paragraph "HDAC6 controls microtubule dynamics ... " seems quite random if not then explain more and link the events.
Thank you for your comment. This paragraph has been amended to reflect the rationale for dual HDAC6/tubulin inhibitor design (lines 482-487, page 14). You can find the modified paragraph below:
“Taking into account that HDAC6 controls microtubule dynamics by deacetylating alpha-tubulin, it can be concluded that the combination of HDAC6 inhibitors and microtubule stabilizing drugs could exert synergistic anticancer effects[94,95]. Considering the results of previous study which has already demonstrated synergistic effects between paclitaxel and the selective HDAC6 inhibitor (citarinostat) [36], there is a rational case for the development of dual-targeting inhibitors”.
- Overall, I feel that the summarized introduction must also include some introduction about systems pharmacological approach which will link the overall story in more interesting way.
Thank you for your suggestion. A schematic representation of the molecular relationships of HDAC6 and other targets can now be found in the manuscript in Figure 3, as well as detailed explanation of it in the Introduction section (lines 118-137, page 4).
- Some of the references, the authors should include in the main text are here:
Thank you for your comment. Regarding the content of this review, we have included this reference in the main text: https://www.mdpi.com/2073-4409/11/24/4121 (References 54, line 150, page 4, section Multi-target therapy as an approach in cancer treatment)
Kind regards

Round 2
Reviewer 2 Report
Comments and Suggestions for Authors
The authors have addressed all my concerns. I recommend accepting this manuscript in current form.
Author Response
Dear reviewer,
Thank you for your great support to improve quality of our manuscript.
Kind regards
Reviewer 3 Report
Comments and Suggestions for Authors
Authors kindly responded to all suggestions and comments.
Just compound structures at (Figure 2II,III) need revision as per "ACS document object settings"
Author Response
Dear reviewer,
Thank you for your great support to improve quality of our manuscript.
- Compound structures at (Figure 2II,III) need revision as per "ACS document object settings"
Thank you for your comment. Figure 2 (2-II and 2-III) is now improved by using “ACS document object setting” (Page 3, Figure 2).
Kind regards